# TEXT GENERATION BY LEARNING FROM DEMONSTRATIONS

**Richard Yuanzhe Pang** [1]
yzpang@nyu.edu

**He He** [1,2]
hehe@cs.nyu.edu

[1] Courant Institute of Mathematical Sciences, New York University, New York, NY 10011, USA
[2] Center for Data Science, New York University, New York, NY 10011, USA

## ABSTRACT

Current approaches to text generation largely rely on autoregressive models and maximum likelihood estimation. This paradigm leads to (i) diverse but low-quality samples due to mismatched learning objective and evaluation metric (likelihood vs. quality) and (ii) exposure bias due to mismatched history distributions (gold vs. model-generated). To alleviate these problems, we frame text generation as an *offline* reinforcement learning (RL) problem with expert demonstrations (i.e., the reference), where the goal is to maximize quality given model-generated histories. We propose GOLD (generation by off-policy learning from demonstrations): an easy-to-optimize algorithm that learns from the demonstrations by importance weighting. Intuitively, GOLD upweights confident tokens and downweights unconfident ones in the reference during training, avoiding optimization issues faced by prior RL approaches that rely on online data collection. According to both automatic and human evaluation, models trained by GOLD outperform those trained by MLE and policy gradient on summarization, question generation, and machine translation. Further, our models are less sensitive to decoding algorithms and alleviate exposure bias.

## 1 INTRODUCTION

A dominant approach to text generation is to use autoregressive models learned by maximum likelihood estimation (MLE) on supervised data. However, this approach introduces two well-known discrepancies between training and evaluation objectives that lead to undesired generations. First, the training loss is negative log-likelihood, whereas the evaluation is based on human judgment of the output quality. Under model misspecification, MLE tends to over-generalize, assigning large probability mass to both high-quality and low-quality sequences (Huszár, 2015; Simon et al., 2019). Therefore, in practice, we must carefully select the decoding algorithms to produce high-quality outputs.

Second, during training, the autoregressive model conditions on the gold history/prefix; however, at inference time it conditions on model-generated history. This is known as the exposure bias problem (Ranzato et al., 2016; Bengio et al., 2015). In the worst case, one incorrect prediction can produce a low-probability prefix under the gold data distribution, and errors compound in each of the following steps (Ross et al., 2011). In practice, prior work has observed problems such as repetition and hallucination partly due to exposure bias (Holtzman et al., 2020; Wang & Sennrich, 2020).

We aim to bridge the gap between training and evaluation in this paper. To match training and evaluation objectives, ideally we should maximize output quality given model-generated histories. This corresponds to the reinforcement learning (RL) objective: maximizing the expected reward (quality) over trajectories (sequences) induced by the policy (model). However, optimizing this objective is notoriously difficult. Prior RL approaches mainly focus on fine-tuning a learned model to optimize sequence-level metrics such as BLEU (Papineni et al., 2002), but empirically it remains unclear if RL is beneficial to text generation (Wu et al., 2018; Choshen et al., 2020). Note that many challenges in RL arise from exploring an exponentially large space of sequences, with sparse rewards only on those close to the reference. We thus propose to learn from only the reference sequences without interaction (i.e., the offline setting). Specifically, we use off-policy policy gradient

with importance weighting (Hastings, 1970; Hachiya et al., 2009; Parshakova et al., 2019), where training examples with higher probability under the model are weighted higher. Further, our reward functions approximate human judgment of the output quality by estimating how likely a human would have generated a sequence. We call our algorithm GOLD (Generation by Off-policy Learning from Demonstrations).

Results on news summarization, question generation, and machine translation show that GOLD leads to better model performance than MLE and RL fine-tuning by both task metrics and human-rated quality. Further, our analysis shows that GOLD learns high-precision models that are less sensitive to decoding algorithms. In addition, it alleviates exposure bias: the output quality does not degrade much as generation length increases.

## 2 FROM MLE TO RL FRAMEWORK

**MLE training.** Given a context $x$ such as a document, we want to generate a sequence of tokens $\boldsymbol{y} = (y_0, \ldots, y_T)$, where $y_i$ comes from a vocabulary $\mathcal{V}$. The generator is modeled by a conditional probability distribution parametrized by $\theta$: $p_\theta(\boldsymbol{y} \mid \boldsymbol{x}) = \prod_{t=0}^{T} p_\theta(y_t \mid \boldsymbol{y}_{0:t-1}, \boldsymbol{x})$, where $\boldsymbol{y}_{0:t-1}$ denotes the prefix $y_0, \ldots, y_{t-1}$. Let $p_{\text{human}}(\boldsymbol{y} \mid \boldsymbol{x})$ denote the data-generating distribution. Using MLE, the loss function is

$$\mathcal{L}(\theta) = -\mathbb{E}_{\boldsymbol{y} \sim p_{\text{human}}} \left[ \sum_{t=0}^{T} \log p_\theta(y_t \mid \boldsymbol{y}_{0:t-1}, \boldsymbol{x}) \right]. \tag{1}$$

At inference time, we generate tokens sequentially according to $p_\theta$.

**Evaluation.** In practice, the quality of an output often relies on task-specific metrics such as fluency, correctness, and interestingness. Here for generality we consider *perceptual quality* (Huszár, 2015; Hashimoto et al., 2019) which measures how likely a human would have generated the output given the context, i.e., $p_{\text{human}}(\boldsymbol{y} \mid \boldsymbol{x})$. Thus the evaluation metric is

$$\mathbb{E}_{\boldsymbol{y} \sim p_\theta} \left[ \sum_{t=0}^{T} \log p_{\text{human}}(y_t \mid \boldsymbol{y}_{0:t-1}, \boldsymbol{x}) \right]. \tag{2}$$

Comparing (1) and (2), we see that the training objective encourages *high recall*: the model must put probability mass on all human-generated sequences. In contrast, the evaluation metric encourages *high precision*: all outputs from the model must be of high quality. Unfortunately, directly optimizing the evaluation metric is impossible because $p_{\text{human}}$ is unknown and the expectation is difficult to estimate. We therefore develop a training objective that closely approximates (2) in the RL framework.

**RL formulation.** Let's consider generation as a sequential decision-making process. At each time step $t$, the policy $\pi_\theta$ takes an action $a_t \in \mathcal{V}$, transits to the next state $s_{t+1} = (\boldsymbol{y}_{0:t}, \boldsymbol{x})$, and receives a reward $r_t$. The policy corresponds to the generation model: $\pi_\theta(a_t \mid s_t) = p_\theta(a_t \mid \boldsymbol{y}_{0:t-1}, \boldsymbol{x})$. We can thus represent a sequence as a trajectory $\tau = (s_0, a_0, r_0, \ldots, s_T, a_T, r_T)$. The set of trajectories derived from the training data is called *demonstrations* which show the desired behavior of a policy. The RL objective is to maximize $J(\theta) = \mathbb{E}_{\tau \sim \pi_\theta} \left[ \sum_{t=0}^{T} \gamma^t r_t \right]$, where $\gamma \in (0, 1]$ is the discount factor, and $\pi_\theta(\tau)$ denotes the distribution of $\tau$ induced by $\pi_\theta$. If we knew oracle rewards $r_t = p_{\text{human}}(a_t \mid s_t)$, then this objective would be exactly the evaluation metric we want to optimize. Next, we describe how to optimize $J(\theta)$ with reward functions that approximate $p_{\text{human}}$.

## 3 APPROACH

### 3.1 OFF-POLICY POLICY GRADIENT

**Policy gradient.** A straightforward way to optimize $J(\theta)$ is policy gradient (PG) (Williams, 1992; Sutton et al., 2000). The gradient is given by

$$\nabla_\theta J(\theta) = \mathbb{E}_{\tau \sim \pi_\theta} \left[ \sum_t \nabla_\theta \log \pi_\theta(a_t \mid s_t) \hat{Q}(s_t, a_t) \right], \tag{3}$$

where $\hat{Q}(s_t, a_t) = \sum_{t'=t}^{T} \gamma^{t'-t} r_{t'}$ is the estimated return from state $s_t$. The expectation is estimated by Monte Carlo samples from $\pi_\theta$. In text generation, the return $\hat{Q}(s_t, a_t)$ is often a sequence-level reward such as BLEU. In practice, the policy is likely to get stuck in a region of zero reward during training, generating gibberish without receiving any learning signal (Li et al., 2018; Keneshloo et al., 2019). A common remedy is to initialize the policy with the MLE solution and/or interleave with MLE gradient update during PG. However, this would bias the parameters towards the MLE solution, thus often leads to marginal gains in practice (Wu et al., 2018; Choshen et al., 2020).

**Offline learning.** To avoid zero-reward regions, we would like to *reduce interaction with the environment and stay close to the demonstrated trajectories*. In the extreme case, the policy is learned solely from the static demonstrations without additional interaction with the environment, which is referred to as the offline setting. While it is in general a more challenging problem, we argue that the offline setting is appropriate for text generation (Serban et al., 2017; Jaques et al., 2019). First, the environment dynamics is known: once a token is generated, we deterministically transition to the next state with the additional token appended to the prefix; no interaction is needed to learn the environment. Second, while exploration may lead to high-quality sequences different from the reference, we lack a good reward function to identify them (Novikova et al., 2017; Aharoni & Goldberg, 2018; Clark et al., 2019). Therefore, the benefit of exploration in text generation is limited.

In the offline setting, we cannot estimate the expected return of $\pi_\theta$ by sampling trajectories from it, and must use trajectories from a different *behavioral policy* $\pi_b$, known as *off-policy* learning in RL. A common technique to estimate expectations under one distribution $\pi_\theta$ given samples from a different distribution $\pi_b$ is importance sampling, which leads to the following unbiased estimator of the gradient (Precup et al., 2000):

$$\mathbb{E}_{\tau \sim \pi_b} \left[ \sum_t w_t \nabla_\theta \log \pi_\theta(a_t \mid s_t) \hat{Q}(s_t, a_t) \right],$$

with importance weights $w_t = \prod_{t'=0}^{t} \frac{\pi_\theta(a_{t'} \mid s_{t'})}{\pi_b(a_{t'} \mid s_{t'})}$.

**Approximations.** Computing the importance weights above requires multiplying per-action importance weight over multiple time steps. In practice, we have found that it is sensitive to optimization hyperparameters and takes longer to converge. Therefore, we use the per-action approximation: $w_t \approx \frac{\pi_\theta(a_t \mid s_t)}{\pi_b(a_t \mid s_t)}$. This corresponds to optimizing the expected return under the off-policy state distribution induced by $\pi_b$ and the on-policy action distribution of $\pi_\theta$. Although this estimator is biased, empirically it has been shown to reduce variance and work reasonably well if $\pi_b$ and $\pi_\theta$ are close (Serban et al., 2017; Levine et al., 2020). Another obstacle is that we do not know $\pi_b$ which produced the demonstrations $\mathcal{D} = \{(\boldsymbol{x}^{(i)}, \boldsymbol{y}^{(i)})\}_{i=1}^{N}$. One option is to estimate $\pi_b$ on $\mathcal{D}$. Here we take a simpler approach that uses the empirical distribution: $\pi_b(\tau) \approx 1/N$ for $\tau \in \mathcal{D}$ and 0 otherwise. As a result, the denominator in $w_t$ is a constant and can be ignored in optimization. Our final approximated gradient has the form:

$$\nabla_\theta J(\theta) \approx \sum_{i=1}^{N} \sum_{t=0}^{T} \pi_\theta(a_t^i \mid s_t^i) \nabla_\theta \log \pi_\theta(a_t^i \mid s_t^i) \hat{Q}(s_t^i, a_t^i), \tag{4}$$

where the superscript $i$ represents the $i$th trajectory. Compared with the MLE gradient: $\sum_{i=1}^{N} \sum_{t=0}^{T} \nabla_\theta \log \pi_\theta(a_t^i \mid s_t^i)$, our gradient (4) upweights actions with high return and actions preferred by the current policy $\pi_\theta$. Intuitively, it encourages the learning algorithm to focus on "easy" examples (high likelihood under the model) which improves precision.

## 3.2 REWARD

Let $R$ be the reward function such that $r_t = R(s_t, a_t)$. To optimize the perceptual quality of a sequence (see (2)), we want $R(s, a)$ to approximate $p_{\text{human}}(a \mid s)$, i.e., how likely humans would have generated $a$ given $s$. In general, it is hard to develop a reliable reward function for text generation tasks because it must work well for a large set of possible generations. In the offline setting, however, we can restrict the domain of $R$ to state-action pairs on the demonstrations. Next, we propose three reward functions.

$\delta$**-reward.** An obvious choice is a sequence-level reward, which considers all demonstrations to be equally good and assigns zero reward to any other outputs. Formally,

$$R_\delta(s_t, a_t) \stackrel{\text{def}}{=} \begin{cases} 1, & \text{if } t = T \text{ and } (s_{0:T}, a_{0:T}) \in \mathcal{D} \\ 0, & \text{otherwise} \end{cases} \tag{5}$$

where a reward of one is received in the terminal state for any trajectory in the demonstrations.

**Estimated $p_{\text{human}}$.** In text generation tasks, an input often has many correct outputs and the reference may be an uncommon output that contains rare words or has complex syntax. To account for different likelihood of the references, we estimate the probability of each reference by minimizing $\text{KL}\,(p_{\text{human}} \| q)$, where $q(a \mid s)$ approximates $p_{\text{human}}(a \mid s)$. This is equivalent to finding the MLE solution (denoted by $p_{\text{MLE}}$).[1] Importantly, $p_{\text{MLE}}$ is a reasonable approximation to $p_{\text{human}}$ when *restricted to the demonstrations*. It is not a good reward function in general, however; it can assign large probability mass to low-quality outputs. Given the estimated perceptual quality $p_{\text{MLE}}(a \mid s)$, we define two reward functions. Our first reward function corresponds to a product of probabilities when summed over the trajectory:

$$R_p(s, a) \stackrel{\text{def}}{=} \log p_{\text{MLE}}(a \mid s). \tag{6}$$

Assuming $\gamma = 1$, the return at time step $t$ is $\hat{Q}_t(s_t, a_t) = \sum_{t'=t}^{T} \log p_{\text{MLE}}(a_t \mid s_t)$. Thus a sequence has high reward only if every word has high likelihood under $p_{\text{MLE}}$. To allow for partial credits even if bad actions are taken at certain steps, we define another reward function corresponding to the sum of probabilities:

$$R_s(s, a) \stackrel{\text{def}}{=} p_{\text{MLE}}(a \mid s). \tag{7}$$

The return $\hat{Q}(s_t, a_t) = \sum_{t'=t}^{T} p_{\text{MLE}}(a_t \mid s_t)$, thus the policy can recover from bad decisions if the subsequent actions receive high reward.

### 3.3 THE GOLD ALGORITHM

Our full algorithm based on off-policy PG is shown in **Algorithm 1**. For importance weights $\pi_\theta(a \mid s)$, to avoid drastic changes, we initialize $\pi_\theta$ with the MLE solution. In addition, we compute the importance weights by a weighting policy $\tilde{\pi}_\theta$ that synchronizes with $\pi_\theta$ periodically so that the weights do not change frequently between updates. We also lower-bound the importance weight by a small number $u$.

Another source of variance comes from policy gradients. Since our return is computed from a sum or product of probabilities ((6) and (7)), we truncate the future trajectory after five steps. We

---

**Algorithm 1:** GOLD

1  $\pi_\theta \leftarrow p_{\text{MLE}}, \tilde{\pi}_\theta \leftarrow p_{\text{MLE}}$
2  **for** $step = 1, 2, \ldots, M$ **do**
3      Sample a minibatch $B = \{(\boldsymbol{x}^i, \boldsymbol{y}^i)\}_{i=1}^{|B|}$
4      **foreach** $(s_t^i, a_t^i)$ **do**
           Compute importance weights
           $\max(u, \tilde{\pi}_\theta)$, and compute returns
           $\hat{Q}(s_t^i, a_t^i) - b$
5      Update $\theta$ by (4) using gradient descent
6      **if** $step \% k = 0$ **then** $\tilde{\pi}_\theta \leftarrow \pi_\theta$
7  **Return:** $\pi_\theta$

---

follow the common practice to subtract a baseline $b$ from the return to reduce variance; moreover, to avoid negative reward on the demonstrations (after subtracting baseline), we lower-bound $p_{\text{MLE}}$ in (6) and (7) by a small number $c$. In practice, GOLD is easy to implement; further, given an existing $p_{\text{MLE}}$, the GOLD-training stage usually takes less time than MLE. The code is available.[2].

## 4 EXPERIMENTS

### 4.1 SETUP

We chose four text generation tasks: (1) **question generation** (**NQG**; Zhou et al., 2017): given a passage and a short span of the passage, the goal is to generate a question that can be answered by the

---

[1] Note that $\text{KL}\,(p_{\text{human}} \| q) = \mathbb{E}_{p_{\text{human}}} \log p_{\text{human}} - \mathbb{E}_{p_{\text{human}}} \log q$, thus minimizing the KL divergence with respect to $q$ is equivalent to the MLE objective.

[2] Code: https://github.com/yzpang/gold-off-policy-text-gen-iclr21

span; (2) **summarization** (**CNN/DM**; Hermann et al., 2015); (3) **extreme summarization** (**XSum**; Narayan et al., 2018): the references are more *abstractive* than CNN/DM summaries; (4) **machine translation** (**IWSLT14 De-En**; Cettolo et al., 2014). See Appendix A.1 for the size and the source of the datasets. We evaluate NQG and summarization by both automatic metrics, i.e., corpus-level BLEU-4 (Papineni et al., 2002) and ROUGE-1/2/L (Lin, 2004) respectively, as well as human ratings.

We experiment with three variants of GOLD: GOLD-$\delta$, GOLD-$p$, GOLD-$s$, which uses the $\delta$-reward and the two estimated rewards ($R_p$ and $R_s$), respectively. Our baseline learning algorithm is standard MLE, and we compare with on-policy RL training using policy gradient in Section 4.3. We describe models for each task at the beginning of Section 4.2.

For GOLD training, we use the baseline $b = -60$ for GOLD-$p$ and $b = 0$ for GOLD-$s$. To lower bound the return such that it is non-negative on demonstrated trajectories, we tune the lower bound $c$ of $p_{\text{MLE}}$ in $\{0, 0.01, 0.05, 0.1\}$ in (6) and (7). Furthermore, to reduce variance for importance weights, we lower bound them by $u \in \{0, 0.1, 0.15, 0.2\}$. All hyperparameters are tuned on the dev set. See Appendix A.3 for more reproduciblility details.

## 4.2 RESULTS AND ANALYSIS

Table 1: BLEU/ROUGE ($\uparrow$) and perplexity ($\downarrow$) using **standard models** on test sets. GOLD achieves better metric scores despite high held-out perplexity. Experiments are run using a fixed random seed (12); attempted three random seeds (1, 12, 123) and all BLEU/R-2 scores are within 0.1 points of the reported. Refer to Table 3 for transformer results.

| | NQG (NQG++ net) | | CNN/DM (pointer generator network) | | | |
|---|---|---|---|---|---|---|
| | BLEU $\uparrow$ | ppl $\downarrow$ | R-1 $\uparrow$ | R-2 $\uparrow$ | R-L $\uparrow$ | ppl $\downarrow$ |
| MLE | 14.23 | **29.25** | 39.00 | 17.10 | 36.07 | **20.11** |
| GOLD-$\delta$ | 14.96 | 110.58 | 39.02 | 17.16 | 35.98 | 133.10 |
| GOLD-$p$ | 15.93 | 148.84 | 39.20 | 17.31 | 36.23 | 143.58 |
| GOLD-$s$ | **16.10** | 158.45 | **39.95** | **17.81** | **36.81** | 29.80 |

Table 2: Dev set results of standard models using different decoding algorithms. $b$: beam size. We report the average of 3 runs for top-$k$ sampling. Models trained by GOLD are less sensitive to decoding algorithms.

| | NQG (BLEU) | | CNN/DM (ROUGE-2) | |
|---|---|---|---|---|
| | MLE | GOLD-$s$ | MLE | GOLD-$s$ |
| greedy | 14.13 | 16.06 | 17.40 | 18.51 |
| beam search ($b = 3$) | 14.19 | 15.84 | 17.65 | 18.44 |
| beam search ($b = 5$) | 14.07 | 15.74 | 17.63 | 18.25 |
| top-$k$ samp. ($k = 5$) | 11.27 | 15.41 | 13.06 | 17.02 |
| top-$k$ samp. ($k = 20$) | 10.08 | 15.38 | 11.23 | 16.57 |

**GOLD improves both standard and transformer models.** Recall that one of our main motivations is that MLE tends to over-generalize under model misspecification, i.e., high recall but low precision. One may wonder whether this problem can be fixed by better modeling. Therefore, we evaluated GOLD with both standard high-performing models and state-of-the-art pretrained model. For standard models, we chose two representative seq2seq-based models, NQG++ (Zhou et al., 2017) and the pointer-generator model (See et al., 2017) for NQG and CNN/DM respectively.[3] Table 1 shows that GOLD is better than MLE in terms of BLEU and ROUGE. In particular, we find that using estimated rewards is superior to the $\delta$-reward, showing the benefits of accounting for varying quality of the references. We thus consider only GOLD-$p$ and GOLD-$s$ in the rest of the experiments. For transformer models (Vaswani et al., 2017), we used the pretrained BART (Lewis et al., 2020) for NQG, CNN/DM, and XSum; we used standard transformer for IWSLT14 De-En. Table 3 shows that GOLD achieves better scores than MLE across all tasks, including near-SOTA on CNN/DM (R-2 95% confidence interval: 21.84-22.33) and good performance on XSum (R-2 95% CI: 22.25-22.92).

We further crowdsourced human evaluation by pairwise comparison[4] between MLE-trained and GOLD-$s$-trained model outputs. Each pair of comparison is repeated three times (by three different workers) and we take the majority answer. For each dataset, the evaluations are done by at least 15 different workers. For NQG, we showed workers the entire input and the questions generated by two models, and we ask workers to select the better one (with a third "tie" option). For summarization,

---

[3]We didn't use standard seq2seq-based models for IWSLT14 and XSum as they are not competitive.

[4]We used Amazon Mechanical Turk on 200 pairs of examples for each of the three tasks. We added qualification tasks with obvious answers and we didn't use any results by workers who failed the qualifications.

Table 3: Results using **transformer models** on test sets. The advantage of GOLD is maintained on advanced models based on transformers and pretraining.

| Objective | NQG (BART) | | CNN/DM (BART) | | | | XSum (BART) | | | | IWSLT14 De-En (Transformer) | |
|---|---|---|---|---|---|---|---|---|---|---|---|---|
| | BLEU ↑ | ppl ↓ | R-1 ↑ | R-2 ↑ | R-L ↑ | ppl ↓ | R-1 ↑ | R-2 ↑ | R-L ↑ | ppl ↓ | BLEU ↑ | ppl ↓ |
| MLE | 20.68 | **5.96** | 44.16 | 21.28 | 40.90 | **5.41** | 45.58 | 22.08 | 37.04 | **5.07** | 34.64 | **5.31** |
| GOLD-$p$ | 21.42 | 10.48 | **45.40** | 22.01 | **42.25** | 11.44 | 45.75 | 22.26 | 37.30 | 6.95 | 35.33 | 7.59 |
| GOLD-$s$ | **21.98** | 7.67 | 44.82 | **22.09** | 41.81 | 9.60 | **45.85** | **22.58** | **37.65** | 6.85 | **35.45** | 6.90 |

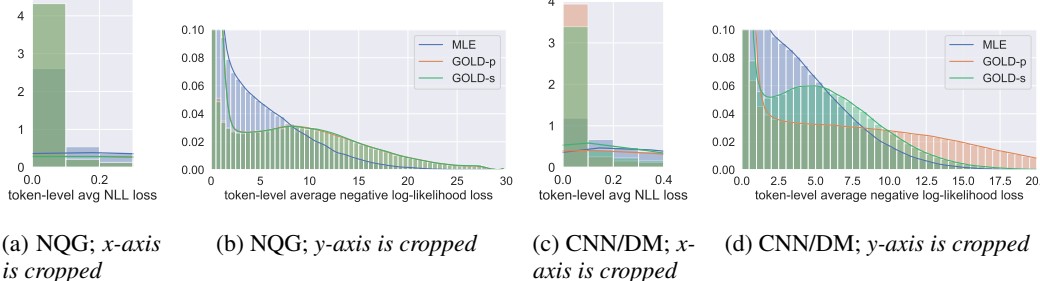

(a) NQG; *x-axis is cropped*  (b) NQG; *y-axis is cropped*  (c) CNN/DM; *x-axis is cropped*  (d) CNN/DM; *y-axis is cropped*

Figure 1: Histograms of token-level NLL loss using standard models on NQG and CNN/DM dev sets. MLE learns high-recall models whose loss distribution is spread out; GOLD learns high-precision models whose loss distribution is concentrated on near-zero losses.

we ask workers to select the generation closer in meaning to the reference without showing the article. More details are in Appendix C. Table 5 shows that workers prefer outputs from models trained by GOLD more often than those trained by MLE.

Table 4: BLEU/ROUGE (↑) on test sets, using standard models finetuned with on-policy objectives. On-policy objectives marginally improve upon both MLE and GOLD baselines. Starred (*) models have MLE baselines >0.1 difference to our MLE R-2. $\delta$R-2: R-2 for the model minus R-2 for the corresponding MLE. PG: standard policy gradient; PPO: proximal policy optimization.

| | reward | NQG | CNN/DM | |
|---|---|---|---|---|
| | | BLEU | R-2 | $\delta$R-2 |
| **Off-policy (this paper)** | | | | |
| MLE | – | 14.23 | 17.10 | – |
| GOLD-$s$ | $R_s$ (Section 3.2); the only *task-independent* reward in this table | **16.10** | **17.81** | **0.71** |
| **On-policy** | | | | |
| MLE+PG | BLEU or R-2 | 14.55 | 17.35 | 0.25 |
| MLE+PPO | human preferences (Ziegler et al., 2019) | – | 17.61 | 0.62 |
| MLE+PG(*) | R-L + saliency + summary-entailment (Pasunuru & Bansal, 2018) | – | 18.00 | 0.67 |
| MLE+PG(*) | question-answering score (Scialom et al., 2019) | – | 17.66 | -0.12 |
| GOLD+PG | BLEU or R-2 | **16.38** | **18.14** | **1.04** |

**GOLD encourages high-precision models.** One interesting observation from Table 1 and Table 3 is that compared to MLE, GOLD leads to much higher held-out perplexities, while achieving better metric scores. Since both are evaluated against the reference, one would expect high perplexity to correlate with low metric scores. To better understand the behavior of GOLD, we examine the distributions of token-level negative log-likelihood (NLL) loss (a monotonic transformation of perplexity) in Figure 1. We see that the loss distribution of GOLD (compared to MLE) concentrates on near-zero losses (Figures 1a and 1c) with a long tail of large losses (Figures 1b and 1d), hence high perplexity. In contrast, MLE has much fewer near-zero losses and fewer large losses, suggesting it tries to generate *all* tokens; i.e., MLE encourages recall, as discussed in Section 2. We conclude

Table 5: Human comparisons on 200 randomly selected test examples for each task. Win: % generations from GOLD-trained BART that are better than from MLE-trained BART, given the same source.

| NQG (BART) | | | CNN/DM (BART) | | | XSum (BART) | | |
|---|---|---|---|---|---|---|---|---|
| win | lose | tied | win | lose | tied | win | lose | tied |
| 38.0 | 28.5 | 33.5 | 37.5 | 24.5 | 38.0 | 35.0 | 21.5 | 43.5 |

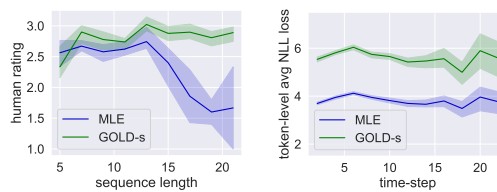

Figure 2: Left: Avg human ratings vs. generation length, on 736 NQG samples. (Colored regions: 95% confidence interval.) Each data point has $\geq$30 annotations. The quality of long generations from MLE-trained model drops heavily, but stays stable across lengths for GOLD-$s$ generations. Right: Avg NLL loss of $t$th token given the *gold* prefix tokens vs. time-step $t$, on NQG dev set. Without exposure bias, NLL loss stays stable across lengths.

that GOLD achieves better metric scores by focusing on easy-to-learn tokens at the expense of lower recall with respect to the reference.

Another advantage of high-precision models is that they do not rely much on decoding algorithms to sample high-quality outputs from the learned distribution. From a RL perspective, the policy already considers future rewards when making local decisions, thus beam search is not necessary. As a result, we see in Table 2 that GOLD achieves similar performance with both argmax decoding and top-$k$ sampling. In contrast, MLE suffers significantly from sampling, which suggests that it learns a high-recall but low-precision model.

**GOLD alleviates exposure bias.** GOLD suffers less from exposure bias because it trains on the state/history distribution induced by the model instead of the reference data. Here, we empirically quantify the exposure bias problem in learned models. If there is exposure bias, then the output quality is expected to degrade as output length increases, as the history is more likely to deviate from the reference distribution with accumulated generation steps. To evaluate quality, we sampled 736 generations of different lengths from standard models trained by both MLE and GOLD on NQG. Given the paragraph, words to query on, and the generated questions, we then asked workers to rate the generations from 1 (worst) to 4 (best). Figure 2 (left) shows that the output quality of the MLE-trained model degrades when the sequence length is over 14 words, whereas the quality of the GOLD-$s$-trained model stays relatively stable across all lengths.[5] Qualitatively, we observe frequent degenerations (Holtzman et al., 2020; Welleck et al., 2020a) including repetitions and hallucinations within a sentence generated by MLE-trained model, as shown in Table 6. In contrast, Figure 2 (right) shows the NLL loss conditioned on gold histories on NQG dev set.[6] We can see that without exposure bias, NLL loss does not vary much as the length increases. Therefore, we conclude that the big performance drop for long generations using MLE is mainly due to exposure bias and GOLD does not suffer from the problem.

Table 6: NQG generations using standard models. Words to query on are bolded. Long generations from MLE-trained model often result in repetition or hallucination. More examples in appendix.

| | |
|---|---|
| Input | that project was entitled **the factory project** to reference andy warhol and to create a factory to completely digitize the collection . |
| MLE | what was the name of the project that was not digitize to digitize ? |
| GOLD | what was the name of the project that was to reference andy warhol ? |
| Input | **braddock** (with george washington as one of his aides) led about 1,500 army troops and provincial militia on an expedition in june 1755 to take fort duquesne . |
| MLE | what was the name of the aid of george washington university ? |
| GOLD | who led about 1,500 army troops and provincial militia on an expedition ? |

---

[5]We also used BLEU as a quality metric and observed similar results, shown in Appendix B.3.

[6]A token prediction accuracy vs. time-step plot, which shows similar trend, is shown in the appendix.

### 4.3 COMPARISON WITH ON-POLICY TRAINING

While offline RL is generally more challenging due to lack of interaction with the environment, we argue that the benefit from interaction is limited in text generation (Section 3.1) and overweighed by the optimization challenges. In this section, we investigate the effect of on-policy training using task metrics as rewards. Specifically, we pre-train the model using MLE and then fine-tune it using PG. To avoid degenerate solutions, we interleave MLE and PG updates evenly during fine-tuning. Similarly, we fine-tune GOLD-initialized models using PG. For on-policy fine-tuning, we use BLEU and ROUGE-2 as rewards for NQG and CNN/DM respectively.[7] Table 4 shows that additional on-policy training improves both MLE and GOLD marginally. However, MLE with PG is still worse than GOLD. Further, one of the best-performing on-policy methods using a similarly competitive pretrained transformer model (Ziegler et al., 2019) also shows limited improvements over supervised baseline on CNN/DM, despite having better reward functions (domain-specific human preference annotations). Overall, the benefit from on-policy training is unclear in our experiments.[8] Please refer to the appendix for more details.

### 4.4 DISCUSSION ON GENERATION DIVERSITY

The objective of GOLD is to produce high-precision text at the cost of recall: There are references that the model cannot generate with high probability, which is reflected by the high held-out perplexity in Table 1 and Table 3. One may wonder what the impact of GOLD on text "diversity" is. This issue warrants more discussion, but for text generation, "diversity" may stand for the following.

**(1) Diversity as in the ability to generate a number of different correct generations given one context.** This is often discussed in the context of mode collapse, which is an important problem for image generation and unconditional text generation (e.g., continuation from a prompt). However, for many conditional NLG tasks, while there are multiple correct outputs, producing one good generation is often sufficient in practice, e.g., question generation, summarization, machine translation, image captioning, text style transfer, and even chit-chat dialogues (unless users expect the bots to say different things in the same context every time). One exception is creative writing tasks where we would like to have multiple novel generations given the same context, e.g., generating from a language model (Caccia et al., 2020). In these cases, GOLD may not be able to provide a variety of high-quality generations given one context, although it would still produce different outputs given different contexts. Another potential failure mode is that in open-ended dialogues, if one common response has large probability under true data distribution, then GOLD may lead to a distribution concentrated on this mode. In this case, additional inductive bias is needed to separate good modes from bad ones, e.g., additional reward on specificity of the response. On the other hand, while MLE-trained models have good recall and we can potentially sample many different outputs with a high temperature, or large $k$ in top-$k$ sampling, or large $p$ in top-$p$ sampling,[9] there are only a few high-quality ones. Our conjecture is that there may not be enough data to cover all modes, and in fact high-likelihood outputs from MLE-trained models are often degenerate (Stahlberg & Byrne, 2019; Cohen & Beck, 2019; Holtzman et al., 2020).

In sum, given the trade-off between diversity and quality, we argue that generating a single high-quality output is a reasonable goal for most conditional text generation tasks, and we leave the question of generating both diverse and high-quality outputs to future work.

**(2) Diversity as in the linguistic complexity of the output, given the input.** First, we compare GOLD and MLE by measuring the complexity of the output using the number of unique n-grams and did not find significant difference. For example, GOLD's number of unique 1/2/3/4/5-grams for XSum (using BART) is 18846/18835/18103/17639/17258, MLE's is 19071/19053/18349/17875/17531, and gold-standard target numbers are 23674/23661/22869/22280/21822. In addition, for question generation and summarization, we measure the complexity of the output by abstractivness, i.e., the proportion of n-gram overlaps between the input and the generation. For XSum (using BART),

---

[7]While rewards in Section 3.2 are useful on *demonstrations*, they are not suitable for the on-policy setting as they cannot differentiate good vs. bad generations on *the entire output space* effectively; e.g., Murray & Chiang (2018); Stahlberg & Byrne (2019) showed that maximizing $p_{\text{MLE}}$ during decoding leads to empty generations.

[8]On a related note, Choshen et al. (2020) also showed in machine translation that even properly tuned on-policy methods may not work well either.

[9]Top-$p$ sampling is another term for nucleus sampling by Holtzman et al. (2020).

the proportion of 1/2/3/4/5-gram overlap for MLE is 0.75/0.27/0.10/0.053/0.031 and for GOLD: 0.73/0.24/0.087/0.039/0.021; the trend mostly holds for NQG and CNN/DM as well. In sum, we conclude that GOLD and MLE are comparable in producing complex or novel outputs.

**(3) Diversity as in the coverage of the true data distribution.** This definition is related to (1). This diversity is the "recall" intuitively, and can be measured by NLL loss or perplexity, which will be sacrificed. In our case, the consequence is that the model tends to ignore difficult gold examples (Figure 1), which in text generation, may sometimes be noise or outliers. Empirically for a large number of text generation tasks, paying less attention to such examples did not cause mode collapse in our case.

## 5 RELATED WORK

**Exposure bias.** In structured prediction, there is a flurry of works addressing exposure bias since Bengio et al. (2015). Most works focus on learning global sequence scores instead of locally normalized scores using either variants of beam search (Wiseman & Rush, 2016; Andor et al., 2016; Goyal et al., 2018) or energy networks (Belanger & McCallum, 2016; Tu et al., 2020). These training algorithms are often complex and costly. Exposure bias is well studied in imitation learning (Daumé et al., 2009; Ross et al., 2011) and learning-to-search has been applied to RNNs to incorporate losses of sequences deviating from references (Leblond et al., 2018), but they require annotations or cost functions on non-reference sequences which may not be available for text generation.

**Objectives beyond MLE.** Policy gradient-based algorithms and their variants have been used extensively in text generation to optimize sequence-level metrics (Ranzato et al., 2016; Shen et al., 2016; Norouzi et al., 2016; Pasunuru & Bansal, 2018). In addition, off-policy RL is commonly used in dialogue where online interaction with users is expensive (Serban et al., 2017; Jaques et al., 2019). The main difference is that we take advantage of the demonstrations and design generic reward functions for generation tasks. There is another line of work using policy gradient to optimize reward from a discriminator that differentiates good vs. bad generations (Yu et al., 2017; Li et al., 2017; Lu et al., 2019). However, these approaches often underperform MLE in practice (Tevet et al., 2019) due to optimization challenges. Recently, a concurrent work, Kang & Hashimoto (2020), proposed truncated log-loss which both optimizes distinguishability and enjoys efficient optimization.

**High-precision text generation.** It is noticed early in neural text generation that MLE tends to produce high-recall models that over-generalize. Previously, high-quality outputs are selected mainly through decoding (e.g., beam search, low-temperature sampling, truncated sampling). Recently, there is an increasing amount of work on discouraging implausible samples during training, e.g., using negative sampling (Welleck et al., 2020b), self-training on high-quality samples (Kedzie & McKeown, 2019), and confidence-oriented decoding with calibration (Tian et al., 2020). In contrast, we tackle the fundamental problem of mismatched objectives and propose a general learning framework.

## 6 CONCLUSION

We provide an efficient algorithm that addresses the two train/test discrepancies in MLE training for text generation: likelihood as learning objective vs. quality as evaluation metric; gold history in training vs. model-generated history in inference. We have demonstrated that off-policy RL is a promising framework for text generation, with matched train/test objectives and optimization advantages like MLE. We believe more advanced off-policy learning techniques (e.g., proximity constraints) can be easily integrated into text generation and further improve performance.

### ACKNOWLEDGEMENTS

The authors thank Kyunghyun Cho, Tatsunori Hashimoto, Graham Neubig, Ethan Perez, Karl Stratos, Clara Vania, and Alex Warstadt (alphabetical order) for helpful discussions, and the anonymous reviewers for helpful feedback. This work was supported by Samsung Advanced Institute of Technology (Next Generation Deep Learning: From Pattern Recognition to AI) and Samsung Research (Improving Deep Learning Using Latent Structure).

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

## A   PRACTICAL SETUP AND IMPLEMENTATION

### A.1   TASKS AND DATASETS

(1) **Natural question generation** (**NQG**; Zhou et al., 2017) based on the SQuAD QA dataset (Rajpurkar et al., 2016): Given a text passage and a short span of the passage, the goal is to generate a question that can be answered by the span. (2) **CNN/DailyMail summarization (CNN/DM)**: Given a piece of news, generate a few sentences of summary. We use the entity-non-anonymized version of CNN/DM dataset, following See et al. (2017). The target summaries tend to be *extractive*, meaning there tends to be heavy text-span overlaps between the source article and the target summary. (3) **Extreme summarization** (**XSum**; Narayan et al., 2018) is based on BBC news. The target summaries are highly *abstractive*. Past extractive strategies that work well for CNN/DM may not work well for XSum. (4) **IWSLT14 German to English machine translation** (**IWSLT14 De-En**; Cettolo et al., 2014) is a popular machine translation benchmark. Machine translation is different from the above three tasks, given that intuitively, the space of high-quality generation is smaller.

**More details on datasets.**   We first provide the number of examples in each dataset. The train/dev/test split for NQG is 86229/8913/8919; the split for CNN/DM is 287227/13368/11490; the split for XSum is 204045/11332/11334; the split for IWSLT14 De-En is 160239/7283/6750.

To download and preprocess the NQG data, we follow the following instructions: `https://github.com/clovaai/FocusSeq2Seq`; to download and preprocess the summarization data, we follow the following instructions: `https://github.com/pytorch/fairseq/blob/master/examples/bart/README.summarization.md`; to download and preprocess the IWSLT14 De-En data, we follow the following instructions: `https://github.com/pytorch/fairseq/tree/master/examples/translation`. More information can be found in our codebase.

### A.2   MODEL ARCHITECTURES

We use two sets of architectures for our experiments.

**Standard architectures.**   For NQG, we use the model NQG++ (Zhou et al., 2017), a seq2seq-with-attention model based on GRU (Cho et al., 2014), and for summarization we use pointer generator network (See et al., 2017), a seq2seq-with-attention model based on LSTM (Hochreiter & Schmidhuber, 1997). Specifically, we use 2 layers for both the encoder and the decoder, for both tasks. Other hyperparameters are based on the following implementation: `https://github.com/clovaai/FocusSeq2Seq`.

**Transformer architectures.**   For NQG, CNN/DM, and XSum, we also experiment with one of the top-performing models, BART (Lewis et al., 2020). Our experiments are based on the pretrained BART model provided by original authors[10]: it has 12 encoder layers and 12 decoder layers, and it is pretrained on around 3.3 billion words of Wikipedia articles and books. We use the model to investigate if our methods work with models with stronger capabilities. For IWSLT14 De-En, we use a moderate-size standard transformer architecture (encoder/decoder embedding dimension 512, 4 encoder attention heads, 6 encoder layers, 4 decoder attention heads, 6 decoder layers), a top-performing architecture in machine translation.

### A.3   MORE ON REPRODUCIBILITY

The codebase is released. The link to the code is posted on the following website: `yzpang.me`.

**Hyperparameters and training details on standard architectures.**   This paragraph corresponds to results in Table 1. We use a learning rate of 5e-4. For NQG, we use a batch size of 32; for CNN/DM we use a batch size of 16. We train using a single Nvidia GTX 1080 Ti (memory: 12 GB) GPU.

---

[10]https://github.com/pytorch/fairseq/tree/master/examples; pretrained by corrupting the original document and optimized with respect to the reconstruction loss between the original document and the decoder output.

As discussed in Section 3.3 and Section 4.1, we tune the lower bound of $p_{\text{MLE}}$ in $\{0, 0.01, 0.05, 0.1\}$. For NQG models, the lower bound of 0.1 produces best performance. For CNN/DM using GOLD-$p$, the lower bound is 0.01; for CNN/DM using GOLD-$s$, the lower bound is 0.

Recall that as discussed in Section 3.3, the weighting policy $\tilde{\pi}_\theta$ synchronizes with actual policy $\pi_\theta$ once every $k$ steps so as to stabilize training. We tune $k \in \{1500, 2691\}$ (where 2691 steps corresponds to 1 epoch) for NQG and found that $k = 1500$ works better for all NQG models. We tune $k \in \{1500, 3000, 5000\}$ for CNN/DM; we found that $k = 1500$ works best for GOLD-$\delta$ and GOLD-$p$, and $k = 5000$ works best for GOLD-$s$. Note that in practice, we do not observe big gaps when using other $k$'s in the set. For standard models, implementation is based on Cho et al. (2019). In all experiments, we evaluate once every epoch, and we do validation on the entire dev set, using task-specific metrics (BLEU/ROUGE-2), following Cho et al. (2019) and standard practice in machine translation.

**Hyperparameters and training details on transformer models.** This paragraph corresponds to results in Table 3. For transformer models, we use Nvidia P40 GPUs (memory: 24 GB each). For NQG, CNN/DM, and XSum based on BART, we use 4 GPUs to train. For IWSLT14 De-En, we use 1 GPU. Note that fairseq defines batch size in terms of number of tokens instead of number of sequences. For NQG, we use 512 tokens as batch size (for each of the four GPUs); for CNN/DM and XSum, we use 1024 tokens as batch size (for each of the four GPUs); for IWSLT14 De-En, we use 4096 tokens as batch size.

We use a learning rate of 2e-5 for NQG, CNN/DM, and XSum; 3e-4 for IWSLT14 De-En.

Recall that as discussed in Section 3.3, the weighting policy $\tilde{\pi}_\theta$ synchronizes with actual policy $\pi_\theta$ once every $k$ steps so as to stabilize training. Here, $k = 1000$ for NQG; $k = 5000$ for CNN/DM, XSum, IWSLT14 De-En. As discussed in Section 3.3 and Section 4.1, the lower bound of $p_{\text{MLE}}$ is set to be 0.01 for GOLD-$p$ and 0.1 for GOLD-$s$. For all other parameters that are not specific to GOLD, we use the default fairseq summarization parameters (which can be found through footnote 10).

For hyperparameter $u$ as discussed in Section 4.1, for NQG and CNN/DM, $u = 0.1$; for XSum, $u = 0.15$; for IWSLT14 De-EN, $u = 0.2$.

As indicated, the hyperparameters were only tuned in a small set of possible values. More careful tuning may result in slightly better performances.

**Number of parameters in each model.** For standard models, we use NQG++ for NQG, and it has 10372565 parameters. We use pointer generator for CNN/DM, and it has 19965705 parameters. For transformer models, the BART model for NQG, CNN/DM, and XSum all have 406290432 parameters; the transformer model used for IWSLT14 De-En has 39469056 parameters.

**Average runtime.** For standard models, based on the above models and the computing infrastructures, each epoch of NQG takes around 10 minutes to train and achieves best performance within 20 epochs. Each epoch of CNN/DM takes about 2 hours to train and achieves best performance within 15 epochs. For transformer models, each epoch of NQG takes around 5 minutes to train and achieves best dev performance within 5 epochs; each epoch of CNN/DM takes around 11 hours to train and achieves best dev performances within 5 epochs; each epoch of XSum takes around 8 hours to train; each epoch of IWSLT14 De-En takes around 3 minutes to train and achieves best performances within 100 epochs (as expected, given the large batch size[11]). Note that our transformer models are trained on P40s given hardware constraints; if the transformer models are trained on V100 GPUs, for example, the training time per epoch will likely be much shorter.

## A.4    More Discussion on Approximations

Recall that we truncated the future trajectory after five steps. In other words, the number of current+future steps is upper-bounded at six. Effectively, we are using a discount factor of 0.83.[12]

---

[11]We use 4096 tokens (which corresponds to hundreds of sentences) as batch size for IWSLT14 De-En.

[12]Note that $1 + \gamma + \ldots + \gamma^T \approx \frac{1}{1-\gamma} = 6$ when $\gamma = \frac{5}{6}$.

Given that the $Q$-value corresponds to future return, we attempted using different strategies. (1) Using the entire future trajectory, and (2) using a fixed number of future steps. We attempted (1) on NQG using the standard models (tuned discount factor in $\{1, 0.9, 0.8, 0.7, 0.5\}$) and found that $\{0.8, 0.7\}$ usually performs best, resulting in similar performance but longer training time, compared to the current 5-future-step approach. We attempted (2) using the number of future steps in $\{1, 2, 3, 5, 7, 10\}$ and found that using $\{5, 7, 10\}$ leads to similar results, which are slightly better compared to $\{1, 2, 3\}$. One benefit is that given a fixed number of future steps, we found that using easy-to-tune constant baselines work well, and training time is also much shorter.

### A.5 DETAILS ON ON-POLICY EXPERIMENTS

For the MLE+PG baseline, we used the REINFORCE algorithm with sequence-level rewards (BLEU for NQG and ROUGE-2 for summarization). We attempted two versions of the baselines: (i) constant baselines searched in $\{0, 0.01, 0.05, 0.1, 0.15\}$ for BLEU (NQG) and ROUGE-2 (CNN/DM), as well as (ii) baselines computed by the average BLEU/ROUGE-2 over the last 100 steps, minus $\{0, 0.05\}$.

In terms of training warmup choices, We tried two versions of the training algorithm. (a) We initialized with MLE and trained with PG losses interpolated with MLE losses, given we found that the training process would become very unstable without interpolation. (b) We also attempted the following: we intialized the model at random and used MIXER (Ranzato et al., 2016). However, we failed to find improvements compared to (a), under our architecture. A relevant work Choshen et al. (2020) showed that properly tuned on-policy RL may not work for text generation in some cases.

We also tried MIXER with the learned baseline for NQG, which is estimated by a simple linear regressor that takes RNN hidden states as inputs, according to Ranzato et al. (2016). After some tuning, we achieved only slight improvements in NQG (BLEU 14.71). One advantage of GOLD is that our algorithm does not rely on learning baselines which could have a big impact on performance of on-policy algorithms; in fact, all baselines are constants in this paper.

Note that for GOLD+PG models, we only attempted constant baselines; better tuning of baselines could potentially lead to stronger performance.

## B MORE ON RESULTS

### B.1 LEAD-3 BASELINES FOR SUMMARIZATION

The lead-3 baseline (using first 3 sentences as summaries) is a popular strong baseline in summarization literature. The ROUGE-1/2/L scores of the lead-3 baselines are as follows: 40.42/17.62/36.67 for CNN/DM; 16.30/1.60/11.95 for XSum. Our performance using transformer models beat these baselines by a large margin.

### B.2 PERFORMANCE WITH TRANSFORMER ARCHITECTURES

We experiment using transformer architectures, as shown in Table 3; we also experiment on two more tasks (compared to using standard architectures): XSum and IWSLT14 De-En. We achieve SOTA/near-SOTA result (according to automatic metrics which have inherent limitations) on CNN/DM: at the time of writing, our results (45.40/22.01/42.25 using GOLD-$p$ or 44.82/22.09/41.81 using GOLD-$s$) are higher than 44.17/21.47/41.11 (PEGASUS; Zhang et al., 2020) and 44.20/21.17/41.30 (ProphetNet; Qi et al., 2020), both slightly higher than BART. Note the PEGASUS CNN/DM result is pretrained on 1.5B news *articles* (around 3.8 terabyte), whereas BART is pretrained on 3.3B words (around 0.16 tetrabyte). Our XSum results are also higher than PEGASUS (45.20/22.06/36.99) trained on Colossal Clean Crawled Corpus (C4; Raffel et al., 2020), but lower than the PEGASUS result using the publicly-unavailable 1.5B-*article* 3.8 terabyte Huge-News (Zhang et al., 2020) as pretrained corpus. We hypothesize that if our models are applied onto their architectures instead of pointer generator networks or BART, we would similarly get non-trivial improvements.

We also achieve 0.81 point of BLEU improvement on IWSLT14 De-En; GOLD-$s$ performs better than the existing approaches that do not use knowledge distillation or data augmentation, as far as the authors are aware.

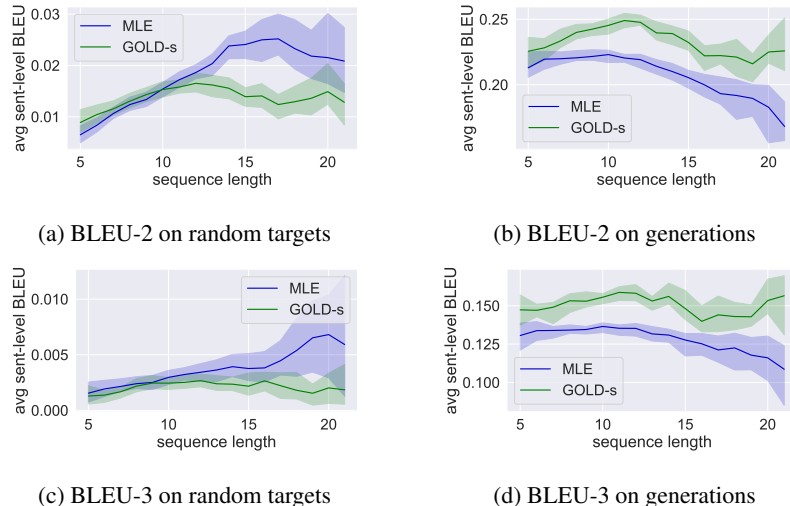

(a) BLEU-2 on random targets

(b) BLEU-2 on generations

(c) BLEU-3 on random targets

(d) BLEU-3 on generations

Figure 3: Exposure bias related figures on NQG dev set. Vertical axis: avg unsmoothed sentence-level BLEU. Horizontal axis: sentence length. The colored regions represent 95% confidence interval obtained using standard bootstrapping. Subfigures (a) and (c) show BLEU on randomly shuffled targets (from dev set); BLEU does not appear to punish long sentences. Note the scale of the vertial axes. Subfigures (b) and (d) show BLEU vs. generation length; BLEU on generations from MLE-trained model decreases by length, but BLEU on generations from GOLD-trained model appears to stay relatively stable.

## B.3 MORE ON EXPOSURE BIAS

**With exposure bias.** Recall that in Section 4.2, we used human evaluation (a score of 1 or 2 or 3 or 4) to approximate the output quality, and we found that the MLE-trained model degrades significantly when the generation length is long, whereas the quality of the GOLD-$s$-trained model stays relatively stable across lengths.

Here, we use BLEU to approximate the quality of NQG generations, and we show that BLEU does not bias toward long sentences. Figure 3 shows the average sentence-level BLEU by sequence length.[13]

Specifically, Figures 3a and 3c show the BLEU on randomly shuffled targets (from dev set), which show that longer sentences do not appear to punish BLEU scores. Figures 3b and 3d show the BLEU by sentence length, on model generations. We see that MLE's BLEU decreases by length but GOLD-$s$'s BLEU appears to stay relatively stable. We thus see some evidence that MLE is generating worse sentences as sentence gets longer.

**If there is no exposure bias.** In the main text, we used the NLL loss vs. length plot to demonstrate that without exposure bias, the loss does not vary much across length, so the MLE performance drop in Figure 2 (left) is mainly due to exposure bias. Here, we provide another way to analyze the case without exposure bias.

Figure 4 shows the token prediction accuracy conditioned on gold histories on NQG dev set. Note that for each example, we let $t_{\boldsymbol{x}} = L_{\boldsymbol{x}} - 5$, where $L_{\boldsymbol{x}}$ is the length of reference sentence $\boldsymbol{x}$. We can see that without exposure bias, prediction accuracy does not vary much as the length increases. Therefore, we conclude that the big performance drop for long generations using MLE is mainly due to exposure bias and GOLD suffers less from the problem.

## B.4 EXAMPLES

Table 7 and Table 8 show the example generations based on the transformer models.

---

[13]We use BLEU-2/3 given that without smoothing, sentence-level BLEU-4 results in large variance.

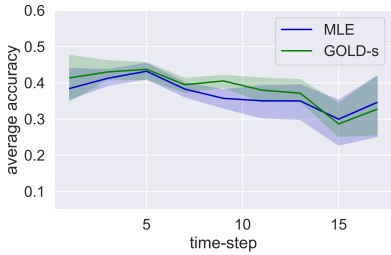

Figure 4: Accuracy of correct predictions of $t$th token given all prefix reference tokens on NQG dev set. Colored regions represents 95% confidence interval obtained using standard bootstrapping. Without exposure bias, token prediction accuracy stays relatively stable across lengths.

| task | objective | example |
|---|---|---|
| NQG | input | Some members of this community emigrated to the United States in the **1890s** . |
| | MLE | when did some members of the portuguese-american community emigrate to the us ? |
| | GOLD-$s$ | when did some members of the community emigrate to the us ? |
| | reference | in what era did some members of this community emigrate to the us ? |
| NQG | input | Competition amongst workers tends to drive down wages due to **the expendable nature of the worker in relation to his or her particular job** . |
| | MLE | what is one of the reasons that causes wages to be lower ? |
| | GOLD-$s$ | why do wages go down when there is competition amongst workers ? |
| | reference | why does competition among workers drive down wages ? |
| NQG | input | **During the mid-eocene** , it is believed that the drainage basin of the Amazon was split along the middle of the continent by the Purus Arch . |
| | MLE | when was the purus arch formed ? |
| | GOLD-$s$ | when was the drainage basin of the amazon split ? |
| | reference | in which point did the drainage basin of the amazon split ? |
| CNN/DM | input | [omitted due to length and copyright issues, but the original news article can be retrieved by searching the reference online] |
| | MLE | There are nearly 5,000 "gems" scattered across the country, ranging from museums to archaeological areas and monuments. Italy boasts the highest number of UNESCO World Heritage sites in the world. Several of which risk crumbling to the ground due to neglect and lack of public resources. |
| | GOLD-$s$ | Italy boasts the highest number of UNESCO World Heritage sites in the world . The Basilica of Assisi, where St. Frances' tomb lies, is badly in need of a restyle . Italy doesn't know how to exploit this treasure, says Francesco Totti . |
| | reference | Italy boasts the highest number of UNESCO World Heritage sites in the world . Italy doesn't know how to exploit treasures, and appears not to care about them, writes Silvia Marchetti . |
| CNN/DM | input | [omitted due to length and copyright issues, but the original news article can be retrieved by searching the reference online] |
| | MLE | President Obama has argued with progressive potentate Elizabeth Warren, calling her "wrong" on trade policy. What everyone does next will be critical for the 2016 elections and the future of Democratic politics. Warren has publicly criticized "fast track" trade authority that would allow the White House to negotiate massive, multination trade deals. |
| | GOLD-$s$ | President Obama has argued with the progressive potentate Elizabeth Warren, calling her "wrong" on trade policy . Julian Zelizer: If Hillary Clinton wants to prove she's a real populist, now is her chance to be even more clear about her position on the TPP deal . |
| | reference | Sen. Elizabeth Warren has publicly criticized so-called "fast track" trade authority . Sally Kohn: Why does President Obama call her wrong, and why is Hillary Clinton equivocating? |

Table 7: NQG and CNN/DM examples based on transformer models. For NQG, words to query on are bolded.

| task | objective | example |
| --- | --- | --- |
| XSum | input | [omitted due to length and copyright issues, but the original news article can be retrieved by searching the reference online] |
| | MLE | The Isle of Wight father's decision not to pay a fine for taking his seven-year-old daughter on holiday during term time caused "a huge amount of confusion", a senior MP has said. |
| | GOLD-*s* | The High Court ruling that a father could not be prosecuted for taking his seven-year-old daughter on a term-time holiday to Disney World caused "a huge amount of confusion", MPs have said. |
| | reference | A High Court ruling backing a parent who refused to pay a fine for taking his child on holiday in term time will cause "huge confusion", an MP has said. |
| XSum | input | [omitted due to length and copyright issues, but the original news article can be retrieved by searching the reference online] |
| | MLE | Flood defences at a Denbighshire beach could be strengthened to reduce the risk of them being breached. |
| | GOLD-*s* | A new dune system could be built to protect a Denbighshire beach from flooding. |
| | reference | New sand dunes may be created to reduce the risk of flooding on a beach on the Denbighshire and Flintshire border. |
| XSum | input | [omitted due to length and copyright issues, but the original news article can be retrieved by searching the reference online] |
| | MLE | Fleetwood's League One play-off hopes suffered a blow as they were held to a goalless draw by League One strugglers Doncaster. |
| | GOLD-*s* | Fleetwood and Blackburn played out a goalless draw in League One. |
| | reference | Fleetwood Town dropped into the League One relegation places as they had to settle for a point after a stalemate with Doncaster. |
| IWSLT14 De-En | input | ich hab da so ne kognitive rückkopplung, du hast was projiziert, was du sehen möchtest. |
| | MLE | i've been doing this with cognitive feedback, you've been prospecting what you want to see. |
| | GOLD-*s* | i've got cognitive feedback, you've proved what you want to see. |
| | reference | i have this cognitive feedback, you projected something you want to see. |
| IWSLT14 De-En | input | es sind also alle werkzeuge vorhanden, und die einzige sache, die uns limitiert, ist unsere vorstellungskraft. |
| | MLE | so there are all the tools available, and the only thing that's licensed to us is our imagination . |
| | GOLD-*s* | so there are all the tools there, and the only thing that limited us is our imagination. |
| | reference | so all the tools are out there, and the only thing that limits us is our imagination. |
| IWSLT14 De-En | input | unser organismus hat eine großartige methode erfunden, um solche unangenehmen gefühle wie neid einfach zum verschwinden zu bringen. |
| | MLE | our organism has invented a great way to get such uncomfortable emotions as neither of us to disappear. |
| | GOLD-*s* | our organism invented a great way to make such uncomfortable emotions like envy easy to disappear. |
| | reference | our organism has come up with an excellent method to make unpleasant feelings like envy simply disappear. |

Table 8: XSum and IWSLT14 De-En examples based on transformer models.

## C  HUMAN EVALUATIONS

### C.1  PAIRWISE COMPARISON

Our goal is to enable high-quality generations that do not necessarily result in gold references. Given that corpus-level BLEU/ROUGE score is only a popular *approximation* of generation quality, we first conduct human ratings to confirm the hypothesis that our approaches are generating better sequences. For NQG, for each unit of human evaluation, we present the source paragraph, the words to ask the question on, the question generated by MLE-trained model, as well as the question generated by GOLD-$s$-trained model. We ask the human evaluators the general question: which generated question is better? Figure 5 shows one example interface of pairwise comparisons.

Using NQG dev set, on standard models, of the 183 pairs of comparison we conducted human evaluations on, 42 (23.0%) MLE-questions are better, 81 (44.3%) GOLD-$s$-questions are better, and 60 (32.8%) are tied. We also evaluate on models based on BART, shown in Table 5 in the main text.

For summarization tasks, given that it is infeasible to get high-quality annotations if we let workers read the entire news article[14], we only did the following: given the reference summary, a summary generated from MLE model, and a summary generated from our model, we asked workers to compare which generated summary is closer in meaning to the reference summary. Figure 6 shows one example interface of the mentioned pairwise comparison for summarization. See Table 5 for results.

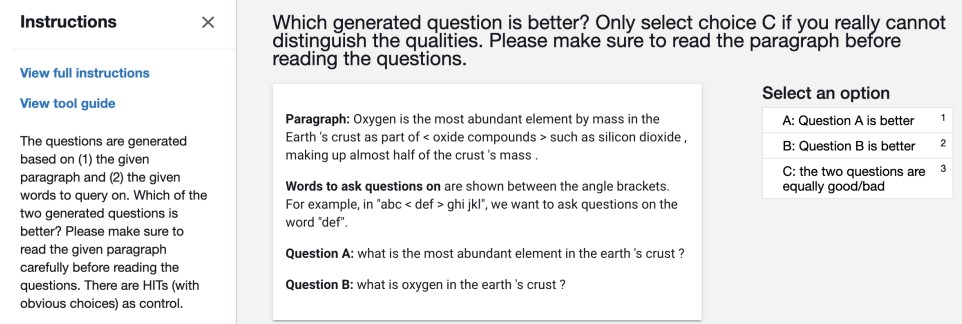

Figure 5: Interface for NQG pairwise comparisons, using Amazon Mechanical Turk.

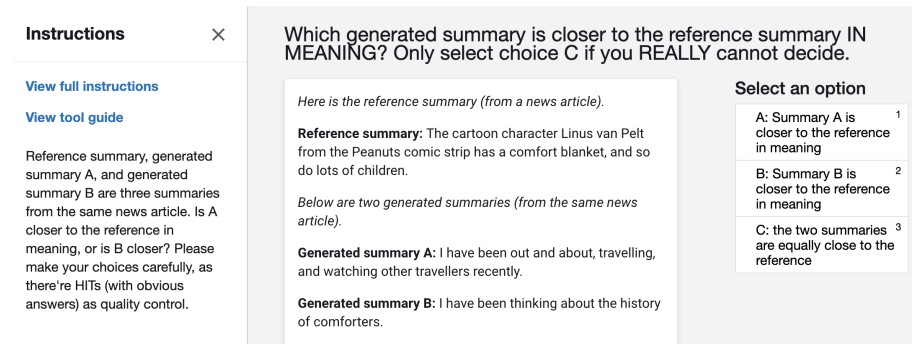

Figure 6: Interface for summarization pairwise comparisons, using Amazon Mechanical Turk.

### C.2  NQG RATING

NQG rating was conducted to examine if longer sentences (generated by MLE-trained model) will result in worse human ratings, and if GOLD alleviates the problem. In Figure 2 (left), to reduce variance, we group length by buckets of two (e.g., $[7, 8]$, $[9, 10]$, $[11, 12]$, etc.). Furthermore, we

---

[14]To obtain high-quality low-variance annotations, we may need to design QA tasks to make sure workers understood the news articles first, given the articles are usually very long.

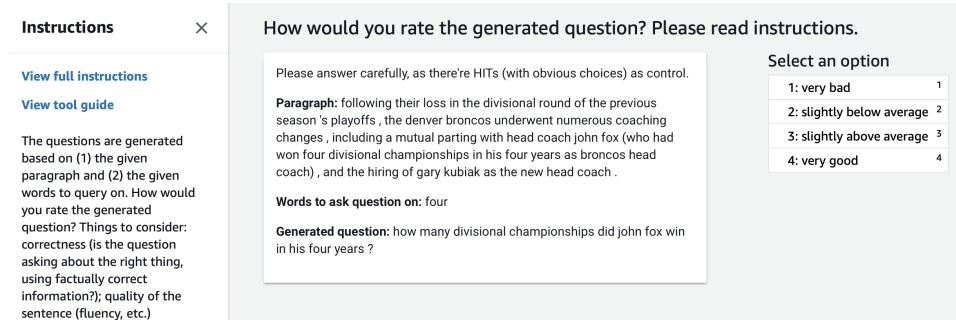

Figure 7: Interface for NQG human ratings, using Amazon Mechanical Turk.

sampled 736 annotations such that each bucket would contain at least 30 sentences (for human evaluation) for each of MLE and GOLD-$s$. We also shown the 95% confidence interval using standard bootstrapping, in Figure 2 (left).

Given the paragraph, words to query on, and the generations, we ask workers to rate the generations. Figure 7 shows an example interface of NQG human ratings. We ask workers to consider both the correctness of the generation (i.e., if the question is asking about the specified words using facts) and the quality of the generation (i.e., if the generation is fluent and coherent). We ask workers to rate from 1 to 4, where 1 means very bad, 2 means slightly below average, 3 means slightly above average, and 4 means very good.

