# OpenReview forum: "Text Generation by Learning from Demonstrations"
_ICLR.cc/2021/Conference — ICLR 2021 Poster_

### Official Review · AnonReviewer4 · 2020-10-27
**An interesting work on high-accuracy text generation**

**Rating:** 7
**Confidence:** 4

**Review:**


#### Summary

This paper introduces a new optimization method for text generation that improves upon directly optimizing MLE. It frames text generation learning as an off-policy reinforcement learning (RL) problem using demonstrations (the training examples). The authors also discuss why off-policy learning is more suitable for text generation than on-policy learning. After simplifications and approximations, the proposed optimization objective comes down to a form that is similar to MLE, but upweighs training examples that are more likely under the learned policy $\pi_\theta$ ("easy" examples) and having higher estimated rewards (considering the future).

Experiments on a variety of text generation tasks and models show that the proposed learning method, GOLD, is able to consistently outperform direct MLE optimization and on-policy learning. The authors also discuss the advantages of GOLD, including an expected higher precision (and lower recall), less performance drop on longer prediction, robustness to decoding algorithms, etc.

#### Strength

- High-accuracy text generation is a valuable research direction, given the well known problems in modern text generation models such as repetitions and hallucinations as mentioned in the paper.
- The proposed learning method is soundly formulated in the manner of off-policy RL. It also makes intuitive sense as upweighing certain training samples based on MLE.
- The experiments justified the general performance improvement the specific advantages brought by the proposed method.

#### Weakness

- As I see, it's not adequately discussed in the paper the cost of sacrificing recall. Intuitively, sacrificing recall in text generation would cause the model to lose output diversity, such as overly favoring the most common sentences like "I don't know" in dialogs. Also, examples are not provided when the proposed method fails to generate good output when other methods can. Although this concern is partly addressed by the good performance of the method in tasks like MT where losing recall can hurt performance for sure, I still feel that more discussions in this line will increase the concreteness.

#### Questions & Suggestions

- Empirically, the proposed learning objective (Eq. 4) has two more factors than MLE: the learned policy likelihood and the estimated reward. It would be interesting to do an ablation study to see how the method performs with each one of them separately, in order to confirm the effect of each factor.
- In Table 2, you reported results on dev set. Why not reporting the test set (or both)?
- In Table 5, you stated that "win" is the percentage of MLE outputs better than GOLD. That would lead to the conclusion that MLE is better than GOLD since "win" is consistently higher than "lose". Should it be the other way around?
- Comparisons to other high-precision text generation methods would be interesting to see (although I understand that most of them are contemporary).

#### Typos

- Section 3.3: "In addition" is used repetitively
- Table 3 caption: "BLUE" -> "BLEU"
- Figure 1(a-d) captions: "zoomed in" is confusing, since it can mean "scaled up". Change to "clipped"?
- Page 7, line 7: missing space after "."

---

> ### Author Response · Authors · 2020-11-18
> **Response**
>
> Thank you for your detailed reviews!
>
> First, we added a post above regarding sacrificing recall and “diversity,” which is especially relevant to the review.
>
> Re: two branching factors for Eq. (4).
> - Great point. Only keeping pi_theta in fact corresponds to GOLD-delta (see Eq. (5) and Table 1), which shows that more fine-grained future return helps.
> - Only keeping Q (but not pi_theta) empirically leads to slightly worse performance on NQG (\~14.6) and CNN/DM (\~17.1 R-2) using standard models. This is not reported in the paper because this variant is hard to be justified theoretically, but the authors do find the degradation of the latter (thus the necessity of pi_theta) intriguing.
>
> Re: Table 2 dev/test set.
> - Our thought process was that we use test set results when we compare results across models, but we use dev set results when we’re investigating the properties of the same model (for example, using different decoding algorithms under the same model). The test set trends for Table 2 are very similar. We’re happy to include the test results if the reviewer finds them standard-practice or finds them helpful.
>
> Re: “win.”
> - Good catch and just fixed! This is a critical typo. It should be the other way around. The corresponding main text is correct, but the caption is wrong.
>
> Re: other typos/suggestions.
> - These are good suggestions! We made the changes based on your comments, and we’ll make sure to proofread more. We will post the updated paper soon.

---

> > ### Comment · AnonReviewer4 · 2020-11-22
> > **Response to rebuttal**
> >
> > Thank you for the response! Most of my questions are addressed (except for comparisons with other high-precision text generation methods, and error analysis, which I'd like to see but not sure if possible). I think the discussions and clarifications make good sense to me. Indeed, I agree that there are a lot of cases where focusing on quality at the cost of "diversity" is desired, which justifies the motivation of this work.

---

> > > ### Author Response · Authors · 2020-11-24
> > > **More discussion**
> > >
> > > Thank you for the reply! Great that the reviewer found our discussion helpful. The below information may not add much to our original response, but we thought we would share.
> > >
> > > In fact, we implemented the recent loss truncation idea by Kang and Hashimoto (2020). They seek to mask out the loss of ~30-90% (a hyperparameter) of the sequences with high loss. Their approach corresponds to GOLD-delta with a binary and sequence-level importance weight. Based on some hyperparameter search, we found that the BLEU for NQG (using standard models in our paper) does not surpass the MLE baseline (may be a similar phenomenon with https://github.com/ddkang/loss_dropper/issues/2). We tried loss_dropping_ratio (the parameter c in the loss truncation paper) in {0.1, 0.2, 0.3, 0.4, 0.5, 0.6, 0.7, 0.8, 0.9}, and we also tried using their algorithm after iterating through {0, 10000, 50000, 100000, i_converge} examples using MLE where i_converge is the number of iterated examples that takes MLE to converge. Granted, the loss truncation idea aims to denoise, minimize distinguishability, increase faithfulness, and preserves (or nearly preserves) diversity; the objective may be a bit different from ours.
> > >
> > > Tian et al. (2020) just updated their WikiBio data-to-text confidence-decoding paper. Kedzie and McKeown (2019) did speech/dialogue-specific data augmentation and self-training on high-quality examples. The two ideas are problem/task-specific, but we will think about how to extend them onto our conditional generation tasks in general. We will also keep an eye on other high-precision generation works!

---

### Official Review · AnonReviewer1 · 2020-10-28

**Rating:** 7
**Confidence:** 4

**Review:**

This paper proposes a method to train generative models of text using reinforcement learning from off-policy demonstrations. This helps solve the problems of exposure bias and mismatched objectives in standard learning schemes such as maximum likelihood estimation and policy gradient optimization on metrics like BLEU. In the proposed method (GOLD), the authors use policy gradient combined with importance weighting to train the model using just the off-policy demonstrations, i.e. human-written text. They experiment with three different reward formulations, and demonstrate improvements over MLE baselines on tasks like summarization and machine translation.

Strengths:
Novel method for training a generative model for text with off-policy learning.
Experiments are well executed and provide interesting insights.
Writing is mostly clear.

Weaknesses:
One or two open questions that could improve the insights provided in this paper (points 3, 5 below)

Questions/comments for authors:
1. It might be worth mentioning somewhere (Introduction?) that this method is for conditional text generation. Although I can see it working for unconditional generation (e.g, language models) as well, as long as human evaluations are performed?
2. Approximation to importance weighting: “In practice, we have found it is sensitive to optimization hyperparameters…” Could you provide a bit more detail/explanation on this? Does it result in lower BLEU scores or does the model not converge at all?
3. It is interesting that the sum of probabilities reward (GOLD-s) performs better than GOLD-p. To me, this seems like maximizing the product of the exponentials of the probabilities (i.e. $\prod_i e^{p_i}$ ) instead of the product of probabilities. This might naturally make the distribution more peaked and hence prevent mass over low quality sequences. So, I’m curious if maximizing the cross-entropy using the exponential distribution (e.g. $\hat{p_i} \propto e^{p_i})$ or adding some auxiliary loss to encourage this would help prevent assigning higher probabilities to low quality sequences as well?
4. “To avoid negative reward… we lower bound $p_{MLE}$”. Isn’t the reward always negative since you take $\log$? I think this is because of the baseline applied later on, but as is, this sentence may confuse readers.
5. As mentioned, the goals of MLE and GOLD seem to be orthogonal (maximize high recall and precision). Why not combine the two objectives? Would that result in a better model?
6. Table 5 caption seems to say MLE generations are preferred to GOLD, but this contradicts the claim in text. Is there a typo somewhere? I would also recommend using more explicit. column names (rather than win, lose) to make it clear which method is preferred.

---

> ### Author Response · Authors · 2020-11-18
> **Response**
>
> Thank you for the comments and questions!
>
> Re: Q1. Yes, this is a really good point! We added a discussion before the related work section. We will post the updated paper soon.
>
> Re: Q2.
> - We put limited information in Appendix A.4. It doesn’t really lower BLEU scores for NQG. We found it helpful to lower-bound/clip each time-step’s probability before multiplying the probabilities into a full-prefix importance weight, but the value of the clip influences the results. Moreover, we need to divide the weights by a denominator given that the prefix length could be very different (for example, divide by d^k for the k-th step importance weight, where d is another hyperparameter to tune; d could be the common choice of empirical probability 1/N or larger). Afterwards, we need to clip the obtained importance weights again in case the weights explode. Different hyperparameters may lead to different results, but only a small number of experiments diverge. To save the trouble, we follow the practice of using a one-step weight. For example, see page 20 and page 21 of Serban et al. (2017; the MILA deep RL chatbot: https://arxiv.org/pdf/1709.02349.pdf); although the importance weight is different from our use case, they also limited to one time-step.
>
> Re: Q3.
> - The correspondence between GOLD-s and the exponential distribution is really interesting! But using MLE will still encourage high recall/coverage, even if we use exponential distribution in learning. Perhaps we can use the exponential distribution in inference? Please correct us if we’re wrong. We’ll think more about this. There are definitely other forms of losses that prevent mass over low quality sequences or outliers, such as the recent loss truncation idea.
>
> Re: Q4.
> - The reviewer is right: we also subtract by the baseline -60. After subtracting, if we don’t lower bound p_MLE, then a small proportion of returns are still negative. Intuitively we didn’t want to punish the model (i.e., take gradients in the opposite direction) from learning gold-standard targets, even if they are noisy/difficult. We fixed the wording a little bit. We will post the updated paper soon.
>
> Re: Q5.
> - We experimented on NQG (standard model) and found that interleaving MLE and GOLD updates will achieve BLEU between MLE and GOLD results. Intuitively, integrating MLE and GOLD might make sense, if we can prevent MLE updates from un-learning GOLD updates, and prevent GOLD updates from un-learning MLE updates.
> - One possibility that’s worth trying and may be promising (in our opinion) is to separate the learning of high-precision regions and high-recall regions. We will definitely think more in this direction and thanks for the suggestion! It’s also possible that it’s a really hard problem to increase both the diversity and precision significantly without major revamping of the models (e.g., perhaps using better pretraining, or adding memory modules in the networks).
>
> Re: Q6. Thanks for catching the typo! Yes, the main text is correct, but the caption is wrong. Just corrected.

---

> > ### Comment · AnonReviewer1 · 2020-11-19
> > **Thanks**
> >
> > Thank you for the detailed responses!
> > Re: MLE with exponential distribution, yes, you're right - I meant during inference since that is what the reward function is helping with. Basically, an interesting baseline might be if you sampled from the model using $e^{p_{MLE}}$ instead of $p_{MLE}$.

---

### Official Review · AnonReviewer3 · 2020-10-28
**Reformulization of Text generation training as an off-policy RL  objective where samples are obtained from the training data**

**Rating:** 7
**Confidence:** 4

**Review:**

This paper formalizes training of conditional text generation models as an off-policy RL objective, specifically, in the limit case where samples are only obtained from the training data. The motivation behind this is that MLE objective optimizes recall -i.e. increasing the prob of all correct sequences that could be generated by humans as an output to a certain input context. While for certain tasks such as MT or summarization it is often sufficient to focus training to generate 1 single correct Translation or summary (see cons: for comments on the effect of this on sample diversity).  Therefore explorations in traditional PG by generating examples from an auto-reg model is unnecessary in this context. Therefore, using importance sampling, PG objective \E_{x \sim \pi_\theta} is modified to \E_{x \sim \pi_{data}} with the incorporation of the importance ratio and given a uniform sample probability of the training examples. As shown in eq(4), The gradient updates from this loss are identical to the standard MLE loss reweighted by the global reward and the current model probability of the training example, enforcing the model's current belief of the training data. This aligns with the previous intuition of enhancing precision on the expense of recall. Given this formulization this work experiments with three types of rewards: a constant reward and 2 MLE based rewards.

Pros:
- formalization of text generation problem as an off-policy RL one, the formalization itself is still interesting, although the fact that in practice no interactions with the environment is needed and all rewards are differentiable.

- The discussion about Precision vs Recall objectives interesting (Although could have been elaborated further discussions see cons 1)
- comparison with strong baselines on 2 tasks and to MLE on 4 tasks QG and summ, Xsum, and MT achieving convincing results on automatic metrics and human evaluation.
- Interesting analysis of exposure bias evaluation of quality w.r.t length compared to MLE models.

Cons:
- Peaking the output distribution of conditional NLG model could enhance the generations precision but on the expense of sample diversity. It is expected that GOLD will more peaked than the one of MLE this can be seen in table2 when increasing the beam size or the top-k sample has little effect on GOLD. This might be suitable for in the case of conditional seq2seq problems,
such as Machine Translation, where focusing on a single good output for a given input makes sense,
but is clearly in-adapted when focusing on language models, or paraphrasing applications where sample diversity is a requirement.
- the choice of constant baselines -60 and 0 is not justified, how did you select such values and why the choice of a constant baseline rather than a simple baseline of mean reward?


Questions:
Q1: is the δ reward just a constant reward of value 1? as examples are just coming from the training data.

Q2: Although I like the formalization of training seq2seq models as an off-policy RL problem.  As it could open more frontiers to more exploration of advanced RL training objectives (e.g. on/off policy interpolation ..etc), the case of this work where all demonstrations are from the training data and no interaction with the env. is required makes the RL formulization a bit hard to digest in this context. I was wondering if and easier formulizations could have been achieved for e.g. A formalization that starts from directly optimizing the rewards using standard supervision since they are all differentiable.

Q3: related to Q2, Could you comment on the conceptual differences between for example maximizing the reward R_p (the MLE based reward) under the proposed RL framework using training data as demonstrations and optimizing the MLE directly?

Q4: Since you initialize \pi_\theta with the MLE model, could one say that all that GOLD does is to increase the peakiness (Choshen et al. 2020) of the MLE model achieving higher precision on the expense of sample diversity which is arguably not necessary for some tasks. If that is the case could one achieve similar gains just by adjusting the temperature of the softmax or by letting the MLE loss overfit the training data more (achieving higher validation PPL by maybe better task reward)?

Q5: What is the motivation behind the lower bound u to the importance weight?

Missing Reference:
Off policy, PG has been proposed in (Parshakova et al. 19) for optimizing global sequence rewards in a distributional perspective rather than an optimization perspective as shown in this paper.

Distributional Reinforcement Learning for Energy-Based Sequential Models, Parshakova et al. 19
https://arxiv.org/abs/1912.08517


minor:
- Typo in the caption of table 5 "Win % generations from MLE that are better than GOLD"  should it be the opposite?

---

> ### Author Response · Authors · 2020-11-19
> **Response**
>
> Thank you for the feedback and questions!
>
> First, we added a post above regarding sacrificing recall and “diversity,” which is relevant to “cons” and Q4, so please check it out. We will also post the updated paper soon.
>
> Re: Q1 about delta reward. Yes exactly!
>
> Re: Q2. Based on the motivation of alleviating train/test objective mismatch and history mismatch, we build our models in the RL framework. In text generation, we argue that the off-policy setting is appropriate because we already know the environment dynamics, and exploration only showed limited gains in the past few years, perhaps because of a lack of good reward function. In practice though, the gradient formulation can be interpreted as MLE weighted by a per-token weight (just like regular on-policy policy gradient’s formulation). The reviewer is right. We do think more advanced on+off-policy interpolation could be interesting.
>
> Re: Q3. Similar to the standard policy gradient approaches, after some derivation, one can find that our approach’s gradient at each time-step is simply MLE gradient multiplied by some weight. The weight in our case comprises two things: the importance weight as well as the future returns Q.
>
> Re: Q4.
> - Please refer to the diversity post on this page, which is especially relevant to this question.
> - Re: adjusting temperature in softmax. That’s a good point, but the rank of the tokens doesn’t change with the temperature choices. Based on Table 2 (effect of decoding algorithms), we see that the current popular decoding algorithms won’t make MLE results catch up with GOLD. As for letting MLE overfit more, in fact we followed Cho et al. (EMNLP 2019) and the standard practice of machine translation to do model selection based on highest validation BLEU/ROUGE. In the MLE setting, this checkpoint always roughly corresponds to the lowest validation loss, based on our observation.
>
> Re: Q5. Most of the original importance weights are actually used, so they are not clipped. We made the decision to lower-bound importance weights by a small value because we don’t want to completely ignore/trivialize gold standards in case the corresponding unclipped weight is small, and it’s possible that small probabilities could be inaccurate.
>
> Re: choice of -60 and 0. One way of choosing baselines is to use advantage A(s,a) = E(s,a) - V(s) where V(s) is the future return with expectation taken over the action space (all possible words). Given that we’re using five future steps, if we’re using GOLD-s, then V(s) would be roughly num_steps*1/vocabsize which is about 0. We also experimented with other baselines and it turns out that the performance is not influenced much by the baselines if they don’t deviate too far from the current value. For GOLD-p, we experimented in {-80, -70, -60, -50, -40} (given that V(s) is in this range); all choices work similarly well for GOLD-p.
>
> Re: reference. We were not aware of this paper. Although not on text generation, we do agree that it’s inspiring. Just added and thank you for pointing it out. We will post the updated paper soon.
>
> Re: typo. Thank you for catching it! Yes the main text is correct but the caption is wrong. Just fixed.

---

> > ### Comment · AnonReviewer3 · 2020-11-24
> > **Thanks for the clarifications!**
> >
> > Thanks for answering my questions and including the extra discussion about diversity in the paper. While there might not be an escape from the quality diversity tradeoff, this clarification might highlight this issue more for future works building on top of this one.  The rest of the answers are satisfactory for me.
> >
> > > Regarding: choice of Baselines and  Q5: lower bound u to the importance weight?
> > It would be great to include this information in the paper for transparency
> >
> > > Re: Q3. Similar to the standard policy gradient approaches, after some derivation, one can find that our approach’s gradient at each time-step is simply MLE gradient multiplied by some weight. The weight in our case comprises two things: the importance weight as well as the future returns Q.
> >
> > One future enhancement is to elaborate this "connections with MLE" in the paper. Such connections are useful to build bridges between communities.
> >
> > ps: I understand that the rebuttal period is over soon so no time for this, but this could be made as a camera-ready enhancement if the paper makes it.

---

### Official Review · AnonReviewer2 · 2020-10-28
**Interesting approach, but scope is overstated, significance based on current results still not clear**

**Rating:** 5
**Confidence:** 4

**Review:**

TEXT GENERATION BY LEARNING FROM OFF-POLICY DEMONSTRATIONS

The authors propose an `````"off-policy" approach to training sequence generation models. The approach is based on using ``"behaviour policy'' or demonstration state distributions and corrects for the action distribution, through a local importance weight, which effectively reduces to the current probability under the policy being trained. Reward functions that assign identity reward to demonstrations, and reward according to the (log/linear) probability of the current sequence under the model are considered. Results suggest that the proposed approach (GOLD), can be utilized after MLE training and before on-policy RL training to improve results, but the experiments are in serveral respects inconclusive (see below for further details).

Strengths:

- The paper is well written, and the approach is straightforward and clearly explained (caveats below).
- Several results suggest GOLD leads to better performance over MLE, and boosts the performance of on-policy RL fine-tuning after MLE (see below for limitations wrt this)
-It is interesting that GOLD reduces exposure bias so much, given that only the action distribution is being corrected for.

Current Limitations:

- The approach is pitched as a new approach to sequence training, and is quite dismissive of strong existing work that looks at hybrid ML/RL objectives, when in fact this approach is applied and only makes sense when applied after MLE training (citations missing, see last bullet for some). More specifically, the overall objective (in terms of importance weight, and optionally reward, c.f. equation 4) self-trains in a way that improves precision at the expense of recall, which the authors themselves state 'encourages the learning algorithm to focus on “easy” examples'. This drops modes, as evidence by poor perlexities reported in table 1, and so it is only appropriate as a fine-tuning step after MLE, or a transition step before doing on-policy RL on a sequence metric.
- The significance of the results is not clear. There are several issues with the results. First, important details regarding the (on-policy) policy-gradient methods that are investigated are missing. Does PG mean REINFORCE? REINFORCE with a learned baseline or other baseline? The baselines utilized in ``PG" and PPO will have a major effect on the results. Is there any warmup for these on-policy RL algorithms, in terms of learning rate, or curriculum learning (Ranzato et al., 2016 as cited)? Again, these are crucial to performance. This is important because the paper should really compare GOLD directly to on-policy results (missing), and/or show that GOLD is a useful step that leads to gains when strong on-policy finetuning is done (done, but not convincing, due to missing details on on-policy RL). In addition, the most appropriate techniques to compare to are those cited in the "High Precision Text Generation" subsection in section 6, but these are not compared to: but this is exactly the primary objective of the work!
-The premise of the paper is to overcome exposure bias and attain higher precision, but the exposure bias issue has been shown to be less severe than originally postulated, and diversity in generation is extremely important in many (most) applications (c.f. Caccia, M., Caccia, L., Fedus, W., Larochelle, H., Pineau, J. and Charlin, L., 2018. Language gans falling short. arXiv preprint arXiv:1811.02549.). Furthermore, mode dropping is an element of GAN/RL training that hybrid ML/RL models have striven to overcome (c.f. Norouzi, M., Bengio, S., Jaitly, N., Schuster, M., Wu, Y. and Schuurmans, D., 2016. Reward augmented maximum likelihood for neural structured prediction. In Advances In Neural Information Processing Systems (pp. 1723-1731).; Ding, N. and Soricut, R., 2017. Cold-start reinforcement learning with softmax policy gradient. In Advances in Neural Information Processing Systems (pp. 2817-2826); Sabour, S., Chan, W. and Norouzi, M., 2018. Optimal completion distillation for sequence learning. arXiv preprint arXiv:1810.01398; so many more...)
-As discussed above, the approach is more limited in scope than advertised. This would definitely need to be corrected to consider accepting the paper.

Overall Assessment:

The paper is well written and the approach is straightforward and interesting, but the premise of the approach, and the experiments supporting the signficance of the method need further strengthening. My initial evaluation leans toward rejection, but I look forward to the authors response.

quality:          7/10 (+solid formulation, +experiments on several datasets and models with analysis)
clarity:           6/10 (+well explained, -scope of paper currently overstated, -important experimental details currently missing)
originality:    6/10 (-more std. application of off-policy RL basics to sequence learning, +intuitive choices for reward fns, policy importance weights)
significance: 5/10 (+important problem, -significance of experiments currently not clear, -importance of overcoming exposure bias and sacrificing recall for precision currently not adequately supported/defended)
overall          5/10 (overall evaluation based on current limitations)

Post-rebuttal feedback:

Authors, thank you for your feedback. The additional comentary and results around diversity have strengthened the paper. However, my other concerns, as described below, remain, and so I have not increased my score, but I have indicated that if an effort is made to address these remaining issues, I would recommend that the paper be accepted.

After reading the reviews and the authors responses, I have several remaining concerns.

-First and foremost, the authors have confirmed that the on-policy results that they compare to are using weak baselines to normalize reward (constant, avg. of last 100 steps). Strong context-dependent baselines are known to be crucial to the performance of on-policy methods. The authors attempted to do some experiments with a MIXER variant without a learned baseline (Ranzato et al, 2016) given my concerns, but MIXER without a learned context-dependent baseline is not MIXER! This is serious, as the conclusions stated in the paper cannot be made until the technique is compared properly to on-policy methods (i.e. that at least utilize context dependent baselines... those with learned q functions [e.g. An actor-critic algorithm for sequence prediction, Bahdanau et al, 2016] are often even more effective)

-The authors did not tone down their claims, or criticisms of on-policy methods requiring MLE pre-training in the paper, despite the fact they also initialize with and ML model, and interleave ML updates during training! The tone of the paper is clearly in need of revision, as I discussed in my review.

-This is not a major concern of mine, but it worth mentioning that the novelty of this paper is actually on the low side. This is a standard application of truncated off-policy learning, and the cited paper out of Bengio's lab (Serban et al, 2017) is in the text domain, and describes essentially the same off-policy approach (although this paper is arguably more clearly presented, and more focused and thorough wrt investigating off-policy variations). In addition, as another reviewer mentioned, the connections to and related work that considers Jenson-Shannon and reverse-KL minimization are strong (and interesting), but they are not discussed/referenced at all.

With all that said, for the most part, I actually like the paper. If the language/position of the paper is toned down/updated, and the results are updated to include stronger on-policy baselines (regardless of the outcome), I think that the paper would be above the acceptance threshold.

---

> ### Author Response · Authors · 2020-11-18
> **Response 1/2**
>
> Thank you for your detailed review!
>
> Re: exposure bias.
> - Our stance is that this is still an open problem with diverging claims. As for the recent popular work on exposure bias which is another ICLR 2021 submission (“quantifying exposure bias for open-ended language generation”; https://openreview.net/forum?id=3teh9zI0j4L), we’re not convinced by the experiments to support the claim that the exposure bias is “not large enough” to reduce performance. The reviewers, especially that paper’s AnonReviewer3, summed up the arguments well (together with issues related to the paper’s investigated tasks and decoding algorithms).
> - Ross et al. (2011) studied the exposure bias problem in imitation learning theoretically, and proved its existence and the fact that errors compound quadratically by time-steps. But does it negatively impact performance in text generation? We are leaning toward yes given the recent evidence. We agree with the contemporaneous work “autoregressive knowledge distillation through imitation learning” by Lin et al. (EMNLP 2020) which shows the impact of exposure bias in translation, and in our submission we also found converging empirical evidence that agrees with Lin et al. (2020). Moreover, exposure bias has proved to contribute to hallucination in Wang and Sennrich (ACL 2020).
>
>
> Re: on-policy methods.
> - One argument for GOLD as an off-policy and offline algorithm is that it is almost as efficient as MLE (the GOLD finetuning step would usually take less time than MLE pretraining), and different hyperparameters do not have a very strong influence on results.
> - But we do agree that clearer explanations for on-policy methods are warranted (added to Section A of appendix). The reviewer is right: PG is REINFORCE. In the paper, we tried (i) constant baselines searched in {0, 0.01, 0.05, 0.1, 0.15} for BLEU (NQG) and ROUGE-2 (CNN/DM), as well as (ii) baseline computed by the average BLEU/ROUGE-2 over the last 100 steps, minus {0, 0.05}. However, we didn’t learn baselines. We tried two versions of the training algorithm with warmup: (a) initializing with MLE and training with PG losses interpolated with MLE losses (without interpolation, the models would fail), and (b) initializing at random and used MIXER (Ranzato et al., ICLR 2016). However, after long tuning of (b), we failed to find improvements compared to (a), under our architecture. A relevant work may be the work “on the weakness of RL for NMT” by Choshen et al. (ICLR 2020) which shows that properly tuned on-policy RL may not work for text generation either.
> - Another blocker for on-policy methods are reward functions. Using expensive and accurate human labels (https://arxiv.org/pdf/2009.01325.pdf) seems to have noticeable improvements over baselines, but it is difficult to obtain high-quality reward function for every task, like the Stiennon et al. (2020) paper.
> - To summarize, GOLD trains more efficiently and requires less tuning/tricks. Moreover, not initializing from p_MLE also works (e.g., initializing from a model trained for a few epochs using MLE, instead of until convergence, on NQG and CNN/DM), as long as the future returns Q are based on p_MLE. We were not trying to justify that GOLD is always better than every on-policy method, but trying to justify that off-policy RL for text generation is suitable and works easily and well, and we want to bring this off-policy-RL & text generation bundle to the attention of both NLP and ML research communities.
>
>
> Re: diversity and the scope of the paper. We added a post above regarding sacrificing recall and “diversity.” Please check it out. In addition (and quite related to the aforementioned post), we want to emphasize that Caccia et al. (2020) uses the setting of unconditional generation of texts from a language model where the goal is to produce diverse generations (measured by self-BLEU), instead of many conditional generation tasks where one good answer is enough (e.g., a single good translation or summary suffice in practice). In the added discussion before the related work section (will post updated paper soon), we qualified our scope of the work as the reviewer suggested. We also made sure that the tasks we experimented on would appear in the abstract and introduction.
>
>
> Re: experiments significance. We added in the caption of Tables 1 and 3 that with different random seeds, the resulting metric scores are almost the same. In appendix A.3 (not required to review), more experiment details are given. Re: high-precision works. We argue that our work is motivated by train/test objective mismatch and history mismatch (exposure bias). Kang and Hashimoto (ACL 2020), for example, was motivated from the fact that log loss is not robust to noise and outliers, and they achieve similar BLEU/ROUGE scores, while having better distinguishability. We’ll investigate if the loss truncation approach draws similar conclusions despite coming from a robustness motivation.

---

> ### Author Response · Authors · 2020-11-18
> **Response 2/2**
>
> We added the discussion of some of the above points into the paper. We will post the updated paper soon. We thank the reviewer again for the feedback and discussion!

---

### Official Review · AnonReviewer5 · 2020-11-06
**Emergency review**

**Rating:** 7
**Confidence:** 3

**Review:**

This paper proposes a new reinforcement learning approach to training generative neural models of text.

The paper's core motivation is that optimizing text generation model with the cross-entropy H(p_data, p_model) gives too much leeway to the model to assign weight to low quality outputs. The proposed solution, is to minimize H(p_model, p_data) to ensure high precision. A variety of practical solutions are proposed to achieve this, presented under the lens of reinforcement learning. Experimental results show that the proposed training algorithm, "GOLD", achieves better results in terms of generation metrics on a selection of conditional generation tasks (MT, summarization, etc...), despite a lower perplexity.

Overall the proposed method seems promising, although it relies on a lot of approximations and specific design choices, the effect of which is not always clearly explained or empirically investigated

Pros:
- Clear motivation
- Good results on a variety of tasks
- Clear improvement on long sentences

Cons:
- a lot of optimization "tricks" introducing additional hyper-parameters are only mentioned in passing, and their effect is unclear without ablation studies. Some examples:
  * lower-bounding of the importance weights with u: What effect does the value of u have? with the current value, how many of the importance weights are actually used?
  * Periodic updating of the policy used to compute importance weights: why is this necessary? What effect does the periodicity of the update have on results?
  * Truncation of the future trajectory after 5 steps. Why 5? Why not using multiplicative discounting?

Questions & remarks:
- The "high recall vs high precision" problem mentioned in section 2 is very reminiscent of the inclusive/exclusive behavior of the forward/reverse KL divergence, perhaps this is worth mentioning
- In Section 2 again, on the issue of initializing the model with the MLE, the authors say: "However, this would bias the parameters towards the MLE solution, thus often leads to marginal gains in practice.". I think that this claim should be substantiated via a proper citation or additional experiments.
- In the related work: "Reward Augmented Maximum Likelihood" (Nozouri et al. 2017) is worth citing as well
- In references, some papers are cited as arxiv papers even though they were published at archival conferences. This should be fixed. Examples:
  * "Globally Normalized Transition-Based Neural Networks": ACL 16
  * "Minimum Risk Training for Neural Machine Translation": ACL 16

---

> ### Author Response · Authors · 2020-11-18
> **Response**
>
> Thank you for the feedback and the observations!
>
> Re: periodic synchronization for the policy that corresponds to importance weights.
> - This trick is actually not necessary, but this stabilizes the training as follows. We experimented without the periodic synchronization on NQG and CNN/DM using standard models, and found that for NQG, there is no noticeable impact (on training time, on generations), but for CNN/DM, around half of the runs (using different random seeds, different hyperparameters) would diverge. However, if we use the periodic synchronization, none of our runs (using different random seeds, different hyperparameters) diverges. It’s possible that if our model takes a “bad” step during GOLD training, then using the “bad” importance weight might encourage the model to take a more “incorrect” step. This snowballing effect may lead to total degenerations. We do think that this synchronization trick can be investigated theoretically and thoroughly. Although not exactly the same, there are many works on periodically synchronizing (or slowly changing) the target networks in (deep) Q learning (Ohnishi et al., 2019; Riedmiller., 2005; Lillicrap et al., 2016).
>
> Re: effect of lower bound u.
> - Most of the original importance weights are actually used, so they are not clipped. For example, for CNN/DM using GOLD-s under standard models, after initializing with MLE, ~65% of the weights are larger than u=0.1, and after training GOLD, ~60% of the weights are larger than u=0.1. The numbers are higher for stronger models (transformer-based; also due to much lower perplexity shown in Table 3). We made the decision to lower-bound importance weights by a small value because we don’t want to completely ignore/trivialize gold standards, and it’s possible that small probabilities could be inaccurate. Weight clipping is also standard in trust region methods like proximal policy optimization.
>
> Re: truncation after 5 steps.
> - Given that the Q value corresponds to future return, we attempted using different strategies. (1) Using the entire future trajectory, and (2) using a fixed number of future steps. We attempted (1) on question generation using the standard models (tuned discount factor in {1, 0.9, 0.8, 0.7, 0.5}) and found that {0.8, 0.7} usually performs best, resulting in similar performance but longer training time, compared to the current 5-future-step approach. We attempted (2) using the number of future steps in {1, 2, 3, 5, 7, 10} and found that using {5, 7, 10} leads to similar results, which are slightly better compared to {1, 2, 3}. One benefit is that given a fixed number of future steps, we found that using easy-to-tune constant baselines work well, and training time is also much shorter.
>
> Re: initializing with MLE leads RL to be close to MLE.
> - Wu et al. (2018; https://www.aclweb.org/anthology/D18-1397.pdf) and Choshen et al. (ICLR 2020; https://openreview.net/forum?id=H1eCw3EKvH) are relevant works, especially the latter which said RL in MT is likely to improve performance only where pretrained parameters are close to yielding the correct translation, and gains may be due to effects unrelated to the training signal but from the shape of the distribution. Thanks for pointing this out!
>
> Re: forward/reverse KL <-> precision/recall. Yes this is indeed a nice connection, as the reviewer pointed out. MLE corresponds to minimizing forward-KL. We’ll think about if there are relevant interesting theories to add.
>
> Re: correct versions for citations. Thank you for pointing them out! We re-checked the bibliography list and tried to correct all citations from arxiv to archival venues. We also added RAML as you suggested. We will post the updated version of the paper soon.

---

> > ### Comment · AnonReviewer5 · 2020-11-24
> > **Thank you for the response**
> >
> > I thank the authors for the detailed rebuttal.
> >
> > I think that the discussion of the various tricks is very valuable, and I hope all of it can be included in some form (perhaps in the appendix) in a final version of the paper as it can be quite helpful to researchers trying to reproduce or extend this work.

---

### Author Response · Authors · 2020-11-18
**More response; posted updated paper (first revision)**

Re: high-precision generation sacrificing recall and diversity. This issue warrants more discussion; for “diversity,” we believe that the community usually means at least the following three things (and perhaps more).

(1) Diversity as in the ability to generate a number of different correct generations given one context. We agree that “mode collapse” is an important problem for image generation and unconditional text generation (e.g., continuation from a prompt). However, for many conditional NLG tasks, while there are multiple correct outputs, producing one good generation is often sufficient in practice, e.g., question generation, summarization, machine translation, image captioning, text style transfer, and even chit-chat dialogues (unless users expect the bots to say different things in the same context every time). The exceptions include creative writing tasks where we would like multiple novel generations under the same context, e.g., generating from a language model (Caccia et al., ICLR 2020). In these cases, GOLD may not be able to provide a variety of high-quality generations given one context (although it still produces different outputs given different contexts). Another potential failure mode is that in open-ended dialogues, if one common response (e.g., “I don’t know”) has large probability under true data distribution, then GOLD may lead to a distribution concentrated on this mode. In this case, additional inductive bias (perhaps from dialogue literatures) is needed to separate good modes from bad ones, e.g., additional reward on specificity of the response.

On the other hand, while MLE-trained models have good recall and we can potentially sample many different outputs with a high temperature or large k/p in top-k/p sampling, there are only a few high-quality ones. Our conjecture is that there may not be enough data to cover all modes and in fact high-likelihood outputs from MLE-trained models are often degenerate (Holtzman et al., 2020; Stahlberg and Byrne, 2020; Cohen and Beck, 2019). In sum, given the trade-off between diversity and quality, we argue that generating (a single) high-quality output is a reasonable goal for most conditional text generation tasks, and we leave the question of generating both diverse and high-quality outputs to future work.


(2) Diversity as in the linguistic complexity of the output, given the input. (2.1) We can measure the complexity of the output themselves using the number of unique n-grams. For example, GOLD’s number of unique 1/2/3/4/5-grams for XSum (an abstractive summarization task) is 18846/18835/18103/17639/17258, MLE’s is 19071/19053/18349/17875/17531, and gold-standard target numbers are 23674/23661/22869/22280/21822. GOLD only produces a little fewer unique n-grams than MLE. (2.2) Specifically for question generation and summarization, we can interpret the complexity of the output with respect to input by computing the proportion of n-gram overlaps (i.e., what percentage of n-gram in generations are also in the source paragraph/article)? For example, for XSum (using BART), the proportion of 1/2/3/4/5-gram overlap for MLE is 0.75/0.27/0.10/0.053/0.031 and for GOLD: 0.73/0.24/0.087/0.039/0.021; the trend mostly holds for NQG, CNN/DM; so GOLD copies less from source, although copying may intuitively be the easy way for question generation and summarization.


(3) Diversity as in the coverage of the true data distribution. This is related to (1), but (3) doesn’t necessarily guarantee (1). This is the “recall” intuitively, and can be measured by negative log-likelihood loss or perplexity, which will be sacrificed. In our case, the consequence is that the model would tend to ignore difficult gold examples, which in text generation, may sometimes be noise or outliers. Empirically for most text generation tasks, paying less attention to such examples would not cause the image-generation-like mode collapse which is considered bad.


We will add a similar discussion in the paper, before the related work section. We ~~will post~~ just posted the first revision of the paper (as well as responses to each reviewer).

---

### Decision · Program_Chairs · 2021-01-07
**Final Decision**

**Decision:**

Accept (Poster)

**Comment:**

The paper is well-written, it is clear and concise. The idea of learning to generate text from off-policy demonstrations is interesting. The results experimental results are good. The authors seem to address the concerns raised by the authors during the rebuttal.